# Oxidation of sulphides and rapid weathering in recent landslides

Robert Emberson[1,2], Niels Hovius[1,2], Albert Galy[3], Odin Marc[1,2,]

[1]GFZ Deutsches Geoforschungszentrum, 14473 Potsdam, Germany

[2]Institute of Earth and Environmental Science, University of Potsdam, 14476 Potsdam, Germany

[3]CRPG-CNRS-UL, 54500 Nancy, France

*Correspondence to:* robert.emberson@gfz-potsdam.de

**Abstract**

Linking together the processes of rapid physical erosion to the resultant chemical dissolution of rock is a crucial step in

building an overall deterministic understanding of weathering in mountain belts. Landslides, which are the most volumetrically important geomorphic process at these high rates of erosion, can generate extremely high rates of very localised weathering. To elucidate how this process works we have taken advantage of uniquely intense landsliding, resulting from Typhoon Morakot, in the Taimali river and surrounds in Southern Taiwan. Combining detailed analysis of landslide seepage chemistry with estimates of catchment-by-catchment landslide volumes, we demonstrate that in

this setting the primary role of landslides is to introduce fresh, highly labile mineral phases into the surface weathering environment. There, rapid weathering is driven by the oxidation of pyrite and the resultant sulphuric acid-driven dissolution of primarily carbonate rock. The total dissolved load correlates well with dissolved sulphate – the chief product of this style of weathering – in both landslides and streams draining the area ($R^2$ = 0.841 and 0.929 respectively, p <0.001 in both cases), with solute chemistry in seepage from landslides and catchments affected by significant

landsliding governed by the same weathering reactions. The predominance of coupled carbonate-sulphuric acid driven weathering is the key difference between these sites and previously studied landslides in New Zealand (Emberson et al. 2015), but in both settings increasing volumes of landslides drive greater overall solute concentrations in streams. Bedrock landslides, by excavating deep below saprolite-rock interfaces, create conditions for weathering where all mineral phases in a lithology are initially unweathered within landslide deposits. As a result, the most labile phases

dominate the weathering immediately after mobilisation and during a transient period of depletion. This mode of dissolution can strongly alter the overall output of solutes from catchments and their contribution to global chemical cycles if landslide-derived material is retained in catchments for extended periods after mass wasting.

**1. Introduction**

Bedrock landslides can produce favourable conditions for weathering by funnelling of runoff into deposits of broken rock mass with massive free mineral surface area and modest hydraulic conductivity. It has been shown in the Western Southern Alps (WSA) of New Zealand (Emberson et al. 2015) that this can give rise to strong gradients in dissolved solid concentrations in surface waters on hillslopes, with highest concentrations in seepage from landslides, and that the

rate of landsliding is an important control on the chemistry of the rivers draining the mountain belt. Localized landslide weathering, prone to the spatial and temporal variability of mass wasting, contrasts to common models in which weathering is controlled by steady and distributed erosion of a laterally continuous soil and regolith mantle (Dixon and von Blanckenburg 2012; Hilley et al. 2010; Ferrier et al. 2008; West 2012, Lebedeva et al. 2010, Li et al. 2014). Such models anticipate that at high erosion rates weathering is kinetically limited, so that the link between material supply

and weathering is progressively weakened and finally lost. Landsliding circumvents this by exhuming relatively unweathered bedrock from deeper below the surface (Gabet 2007) and, crucially, fragmenting the mobilized materials so that weathering is locally optimized. Landsliding is the dominant erosion process in orogenic settings where rapid crustal shortening is paired with fast and persistent exhumation (Dadson et al. 2004; Hovius et al. 1997). We expect, therefore, that it may act as a first order control on dissolved solids export from active mountain belts. Here, we ask:

does highly localised landslide weathering persist in different settings with variable lithology and climate so that a more general case for the impact of landslides on weathering can be made, and if so, then what is the underlying mechanism by which weathering is boosted?

To address these questions, we have investigated chemical weathering in the rapidly eroding mountains of Taiwan at the landslide and catchment scale. Our study site has a substrate of typical continental margin sediments, comprising

calcareous sandstones and shales, and containing sulphide minerals (Das et al. 2012). This is a marked contrast with the WSA, where the meta-sedimentary silicate rocks contain only limited amounts of carbonate and very little pyrite (Chamberlain et al. 2005, Koons and Craw 1991). In this study, we exploit the synchronous occurrence of voluminous mass wasting in southern Taiwan due to exceptional rainfall associated with typhoon Morakot in 2009 to eliminate the effects of progressive loss of soluble minerals, allowing a sharp focus on the causes of elevated weathering rates in

recent landslides. Using samples of seepage from recent bedrock landslide deposits and water from mountain streams and rivers paired with catchment-wide estimates of landslide volume, we show that enhanced weathering in landslide deposits results primarily from the rapid reaction of the most labile minerals in the substrate, which are excavated from below the saprolite-bedrock interface by deep-seated slope failure. In the case of our Taiwanese site, the oxidative weathering of highly reactive pyrite and the ensuing rapid sulphuric acid-driven weathering of carbonate dominate the

output from recent landslides and are a first order control on the stream chemistry.

## 2. Field Sites and sampling

The mountains of Taiwan have erosion rates that are amongst the highest on Earth (between 4-6mm/yr (Willett et al. 2003; Fuller et al. 2003), driven by rapid tectonic convergence at the East Asian margin and aided by tropical storm

activity and earthquakes (Dadson et al. 2003). Average annual mass-wasting rates are high (Hovius et al. 2000), but can be pushed to much higher rates by single, exceptional storms or seismic activity (Hovius et al. 2011; Lin et al. 2008; Marc et al. 2015). For example, in 2009 typhoon Morakot caused over 2.7m of rain in 48 hours (Chien and Kuo 2011) on the Southern part of the island, the highest storm totals ever recorded in Taiwan. The typhoon generated more than 22000 landslides (Lin et al. 2011; West et al. 2011), with an average density of 35000 $m^2/km^2$, orders of magnitude more

than under less exceptional meteorological conditions. This presents a rare opportunity to study an extensive population of recent, synchronous landslides, which is important because the rapid weathering in landslides might decay over a decadal time scale to levels indistinguishable from that on surrounding hillslopes (Emberson et al. 2015). Having a constant, known date of initiation for sampled landslides reduces any systematic changes to weathering that would be introduced by variable timing of the mass wasting. Moreover, having a total landslide surface area and volume

dominated strongly by input from one single typhoon makes the solute load of rivers draining catchments with otherwise diverse landslide histories directly comparable.

We sampled water from a variety of sources within the Taimali river catchment (Fig.1b) between February and March 2015. This catchment drains an area of 118km$^2$ in the East flank of the Taiwan Central Range, from the main divide to the Pacific Ocean. It had among the highest catchment landslide densities resulting from typhoon Morakot, and several

subcatchments sustained catastrophic alluviation. As a result, human settlement and activity in the catchment were abandoned inward of 7km from the coast, and at the time of sampling, anthropogenic influence was negligible. In addition to the Taimali main stream, we sampled stream water from 19 subcatchments with areas between 1-10km$^2$, and geothermal spring water. We also sampled seepage from the deposits of 14 landslides. This water has circulated through the landslides and derives from rainfall, and in some cases is likely supplemented by runoff into crests of landslides. We supplemented our landslide seepage samples from the Taimali catchment with seepage from 8 recent landslides elsewhere in Southern Taiwan (Fig.1A), to assess landslide weathering over a wider range of locations.

Lithologically, the Taimali catchment is representative of the Tertiary sedimentary cover that forms a large proportion of the Taiwanese orogen (Central Geological Survey 2000); the slates, shales and sandstones of the Miocene Lushan and Eocene Pilushan formations contain important amounts of sedimentary and diagenetic carbonate and are pyrite bearing throughout (Das, et al. 2012). The sampled landslides outside the Taimali catchment are also rooted in the Pilushan formation, with the exception of two landslides in the Chenyoulan catchment in central West Taiwan, underlain by the Eo-Oligocene Paileng formation of slates and sandstones. Elsewhere in Taiwan the sedimentary carbonate content of the sampled formations is between 0.28-0.66% (Hilton et al. 2014), and the sulphide abundance in the South of Taiwan has been estimated at 0.05-0.27% (Kao et al. 2004). The lithologies of these formations are the dominant sedimentary rocks formed at continental margins with a clastic input, and they are found globally in active mountain belts (Hartmann and Moosdorf 2012).

Sampled landslides vary in size between 7.6x10$^3$-2.8x10$^6$m$^2$, and their deposits cover a range of shapes. The largest landslides in the Taimali catchment initially blocked the main river channel and have since been incised, leaving massive fan terraces, tens of metres above the current river level. Intermediate landslides have built debris cones on hillslopes and at the edges of valley floors, and some smaller landslides have filled debris chutes. All sampled landslides had mobilized bedrock in addition to the overlying regolith and soil mass and any present vegetation. Soils are generally only >1m in local concavities in the landscape (Lee and Ho 2009), but contiguous, dense vegetation on hillslopes that have not been dissected by landsliding suggests sufficient mobile regolith or soil is present to support flora. In the Taimali catchment and at other landslide sample sites, the modal topographic slope is around 26°, with many steeper hillslope segments.

Seepage was found at the base of many landslide deposits, immediately above the interface with underlying bedrock. The flow paths of this seepage could enable fluid interaction with exposed rock in the landslide scar as well as fragmented rock mass in the deposit.  We do not draw distinction between these, as the physical process of bedrock landsliding remains responsible for the exposure of the scar and the production of the deposit. Moreover, it is not possible to distinguish systematically between landslide scar and deposit from available satellite imagery. Landslide seepage may also incorporate deeper groundwater, which could exfiltrate through the scar. We compare sampled seepage with our own measurements of geothermal water as well as with the composition of deep groundwater reported in the literature to qualitatively assess the importance of deeper-sourced water in landslide outflow.

Notably, during the early part of 2015, when we sampled, the South of Taiwan underwent one of the driest periods on record, with an estimated return time of 50-100 years (CWB (Central Weather Bureau) 2016). At this time, the sampled streams were at or close to a baseflow condition, but despite the dry conditions more than half of around 50 deep-seated landslides we visited had persistent seepage. It seems probable that many landslides in Southern Taiwan are never fully 'dry' in the current climate. Landslides in the Chenyoulan catchment were sampled in July and November of 2013, when more normal rainfall conditions prevailed.

Further, it is important to note that the sediment generated by landslides and debris flows does not reside exclusively on the hillslopes from where it was sourced. Significant storage of material over long timespans in larger river channels is common, and is indeed substantial along the Taimali River, with alluvial channel fill occupying about 6% of the total catchment area. This channel fill may be an important weathering reservoir, and to investigate whether this stored sediment affects the observed river chemistry we have mapped the fill deposits along sampled streams.

Since we sampled small seepages and 1st order tributaries, we lack good constraints on the fluxes of solutes. Discharge gauging is not available in the Taimali, as the hydrometric infrastructure on the river was destroyed by channel bed aggradation after typhoon Morakot. Here, we report measured solute concentrations in landslide seepage and stream waters with unknown flow rates. These measurements can't be converted into solute fluxes without major assumptions, and weathering flux estimation is not the purpose of this study. Instead, we aim for a direct comparison of landslide

seepage and river chemistry from a set of lithologically similar streams with identically timed mass wasting but varying populations of landslides.

## 3. Methods

In this section, we report the analytical techniques used to obtain the data- in section 4 summarise these results. In later discussion, we compare measurements of both chemical concentrations and ratios with physical data about landslides and catchments derived from remote sensing. In general, we focus on qualitative links between these parameters, using statistical tests ($R^2$ coefficient of determination, and p-value to test significance of relationships) to support suggestions that parameters are connected.


### 3.1. Analytical Techniques

Samples were collected from source using an HDPE syringe, filtered on site through single-use 0.2µm PES filters into several HDPE bottles, thoroughly rinsed with filtered sample water. Samples for cation analysis were acidified using ultrapure $HNO_3^-$. pH and temperature values were measured in the field at the time of sample collection.

Analysis of cations was carried out with a Varian 720 ICP-OES, using SLRS-5 & USGS M212 as external standards, and GFZ-RW1 as an internal standard and quality control (QC). QC samples were included for every 10 samples to account for drift; no systematic drift was found, with random variability less than 5%. Sample uncertainties were determined from calibration uncertainties, and were always lower than 10% (see Table 1 for element-specific uncertainties). Anion analysis was performed using a Dionex ICS-1100 Ion Chromatograph, using USGS standards

M206 & M212 as external standard and QC. Uncertainties were always less than 10% for each of the major anions ($Cl^-$, $SO_4^{2-}$, and limited $NO_3^-$). Bicarbonate ($HCO_3^-$) was calculated by charge balance, which has been shown to introduce errors of approximately 10% (Galy and France-Lanord 1999). We make the simplifying assumption that the Dissolved inorganic carbon (DIC) is formed only of $HCO_3^-$, which is generally applicable at the range of pH values measured (Zeebe and Wolf-Gladrow 2001). The calcite saturation index of measured waters was determined using the USGS

PHREEQC software package (Pankhurst and Appelo 2013).

### 3.2. Cyclic Input

Solute input from atmospheric sources can be significant in rivers and streams, and is routinely corrected for in many weathering studies. Previous work on the average chemistry of rainfall in different parts of Taiwan, has found that it is

highly dependent on location (e.g. higher pollutant content near the densely populated west coast (Wai et al. 2008) and intensity of rainfall (typhoon rainfall boasts the highest solute concentrations in some cases (Cheng and You 2010)). Using any of these measurements to correct our samples based on the assumption that all present chloride derives from cyclic input (following e.g. Calmels et al. 2011) resulted in clearly erroneous estimates of the proportion of cyclic salts in the water samples (e.g., >100% of dissolved $K^+$ and $SO_4^{2-}$ from cyclic sources). Notably, chloride is a minor

component in all samples, with a median value of 32 μmol/l (minimum 14.6, maximum 175 μmol/l). This is a similar range of values to rainwater measured in other coastal environments (Galloway et al. 1982; Nichol, Harvey, and Boyd 1997), and the overall correction of TDS is at most 10% even with the most concentrated rainfall we have found in published literature (spot samples measured by Calmels et al. (2011) in the Taroko gorge). We note that deep groundwater may also introduce chloride, which could lead to overestimates of cyclic input. Moreover, during the dry

sampling conditions, excess evaporation could lead to larger proportions of cyclic solutes. In view of the minor amounts of chloride in our samples and because of the caveats mentioned above, we elect not to correct our samples for cyclic input. In the discussion, we assess whether this simplifying assumption is valid.

**3.3. Mapping of landslides**

We mapped 576 landslides extant in the Taimali and surrounds prior to typhoon Morakot and those generated during and after the event until 2015 from LandSat imagery (LS7 & LS8, resolution 30 and 15m respectively). Landslides generated during typhoon Morakot were also mapped from a mosaic of Formosat images taken within one month of the typhoon with a resolution of 2m. Landslides were identified manually, based on their geometry, topographic position

and surface properties, as described in recent studies (Marc and Hovius 2015), and digitized on screen in an Arc-GIS environment; scars and deposits were mapped together as a clear distinction was not always possible from the available imagery. Manual mapping prevents the inclusion of other active landscape elements with recent reflectivity changes, for example aggraded riverbeds, which were mapped separately. It also suppresses the possibility of amalgamation of multiple landslides into single mapped polygons, which blights automatic mapping (Marc and Hovius 2015, Li et al.

2014). In our analysis we draw distinction between older landslides and those generated by typhoon Morakot, prompted by our previous work (Emberson et al. 2015) that has shown the chemical output from individual slides is likely to significantly decay over a decadal timescale. It should be noted that the proportion of landslides predating typhoon Morakot is dwarfed by the mass wasting caused by this exceptional typhoon in much of the Taimali catchment. The proportion of the Taimali catchment occupied by landslides increased from 2% to 8% due to typhoon Morakot, while

some subcatchments have upwards of 16% of their area occupied by typhoon-induced landslides. However, in a few subcatchments there is a larger proportion of older landslides, which allows for comparison with landscape units with only negligible prior mass-wasting.

From the mapped landslide areas, we also estimate the volumes of individual slides, using global area-volume relationships (Larsen et al. 2010). We lack direct measurements of the depth to which the landslides we sampled have

scoured. Errors in the mapped landslide area are assumed to be <20%, which are propogated into the total volume estimated, as well as the catchment-wide volume estimates. This follows the approach described in (Emberson et al. 2015), which allows direct comparison of the estimated landslide volumes in Taiwan and the WSA (Figure 4).

In addition, we have quantified the area, modal slope, and the mean and range in elevation of each catchment in an ArcGIS environment, analysing ASTER GDEM V2 data provided by NASA. These data are reported in table 2.

**4. Results: Landslide Seepage and Stream Chemistry**

Concentrations of Total Dissolved Solids and major cations and anions in landslide seepage and stream water in the Taimali catchment and elsewhere in Taiwan are listed in Table 1.

TDS in seepage from sampled landslides in Taiwan ranges between 7080 μmol/l to 23300 μmol/l, while river TDS values lie between 4230 μmol/l and 12700 μmol/l. Thus, the measured total dissolved solids (TDS) in seepage from sampled landslides is consistently higher than in other surface water sources, with the exception of geothermal springs (up to c.121000 μmol/l)

Dissolved Calcium is the major cation in most samples, forming up to 90% (molar) of the cation load in some cases. In some seepage samples this is replaced by Magnesium, with $Mg^{2+}$ as much as 70% of total cations. Sodium makes up much of the remainder of the cationic load, although it only exceeds 15% molar in three seepage samples. Dissolved Silicon (as silicic acid in these pH conditions) never forms more than 10% of TDS. All measured samples are supersaturated with respect to calcite (Table 1). The anion load is always more than 50% Bicarbonate; the remainder is formed of Sulphate, $SO_4^{2-}$, which is always more than 10% and can be as much as 50%. There is significant overlap between landslides sampled within the Taimali river and those in the immediate surrounds (Fig.1A); TWS15-68 is the most notable, with relatively elevated $SO_4^{2-}$ and TDS (4078 and 14932 μmol/l respectively). Seepage from the two sampled landslides in the Chenyoulan catchment share the same overall relationships between individual elements as those sampled in the Taimali catchment, although the measured concentrations are higher than in the majority of Taimali seepage samples.

The concentrations of TDS in the streams of the Taimali catchment – 4230-12700 μmol/l – are systematically lower than in landslide seepage, but high compared with mean global rivers or mountain streams (e.g. Gaillardet et al. 1999, West et al. 2005). $Ca^{2+}$ is again the most significant cation, always forming more than 21% and as much as 37% of TDS; $Na^+$ and $K^+$ are minor components, always less than 7% and 1% of TDS, respectively. $Mg^{2+}$ is quite variable; in most streams it only forms 1-3%, but it constitutes as much as 11% of the cationic load in three of the streams. Silicic acid averages 6% of TDS, In the anion load, $HCO_3^-$ is often the most important component, forming between 20-50% of TDS. Sulphate is between 5 and 31% of TDS and exceeds $HCO_3^-$ in six streams. We also measured trace strontium, the maximum measured value of which is at 30 μmol/l. Lithium in the streams and main river is always below the detection limit of the OES (c.0.1 μmol/l).

**5. Discussion**

**5.1. Landslide weathering**

Seepage from landslides in the Taimali catchment and scattered locations elsewhere in Taiwan has high TDS concentrations when compared with local streams. This supports the hypothesis that landslides are primary seats of rapid weathering in this setting. We have found the same in the WSA of New Zealand (Emberson et al. 2015), and therefore the intense fragmentation of newly exposed rock mass, and the concentration and slow percolation of rain and runoff water in debris accumulated in topographic hollows are likely universal mechanisms promoting landslide

weathering in steep mountain settings where distributed weathering in soils is kinetically limited.

In the WSA, TDS concentrations in landslide seepage are strongly impacted by the dissolution of trace calcite, the abundance of which is likely to be greater at depth (e.g. Brantley et al. 2013). The depth of a landslide scales with its area (Larsen et al. 2010), implying that the amount of calcite as a proportion of the total landslide volume correlates with the landslide size. This correlation will be limited to the range of landslides where a significant proportion of excavated material is from above the depth to which calcite is depleted. By contrast, in the largest landslides, near-surface materials, poor in calcite, form only a small proportion of the deposit, and a correlation with size my no longer hold. Moreover, large landslides may also be subject to reduced diffusion of fluids and oxygen to the base of the deposits; the amount of water entering the slide scales with the area of the slide (Emberson et al. 2015), but the volume increases at a greater rate (Larsen et al. 2010).

In the Taimali catchment, there is no strong correlation between landslide size and TDS; $R^2$ values for correlation between TDS and landslide area (Figure 2), and TDS and landslide volume are 0.11 and 0.09 respectively ($p = 0.62$, $p = 0.70$, respectively). This suggests that here, other factors have an overriding effect on the solute concentration in seepage from landslides. Notably, we find a strong correlation between the TDS and dissolved Sulphate concentration in landslide seepage ($R^2 = 0.88$; $p <0.001$). This is not unexpected, and could arise from dilution or mixing. However, the ratio of calcium to the sum of other cations (sodium, potassium, magnesium and calcium) is correlated with the ratio of dissolved sulphate to the same cations (Figure 3A; $R^2 = 0.64$, $p<0.005$), which suggests a change in weathering regime. A decrease in the fraction of sodium to other cations also correlates with the increasing fraction of sulphate (Figure 3B, $R^2 = 0.63$, $p<0.005$). We attribute these changes to the oxidation of pyrite in the mobilized and fragmented rock mass and an ensuing increase in weathering of carbonates driven by sulphuric acid. Although the sporadic presence of other sulphide minerals, or mineral sulphates, can't be excluded, this has not been documented in Southern Taiwan. Instead, the rapid oxidation of pyrite has previously been observed in the region and elevated TDS concentrations in Taiwan mountain rivers have been viewed in this light (Calmels et al. 2011; Das et al. 2012; Kao et al. 2004). Sulphide oxidation is so prevalent in the larger Kaoping basin (draining the South Western part of Taiwan) that the delivered sulphate is 1.2-1.6% of global fluxes, from a basin only ~0.003% of global drainage (Das et al. 2012), and these prior studies link this explicitly to elevated physical disintegration of rock.

The area and volume of landslides varies over 3 and 4 orders of magnitude respectively; it is unlikely that the distribution of sulphides in the rock mass varies as much over short distance. Instead we suggest that the absence of correlation between size and solute concentration is likely indicative of a limit to advection and diffusion of oxygenated water in landslides, either linked to the size or the unconstrained internal hydrology. The importance of pyrite in landslide weathering in Taiwan is disproportionate with its relatively minor abundance in the overall rockmass (Das et al. 2012; Kao et al. 2004). Instead, its prominent role is determined by its reactivity. By introducing significant quantities of fresh rock material into the surface weathering zone, landslides reset the proportions of individual minerals within this zone to that of the deeper substrate. This differs from weathering linked to soil production, as in many cases the most reactive phases are depleted below the bedrock-regolith interface (e.g. Brantley et al. 2013, Drake et al. 2009).

The majority of the high TDS in landslide seepage is accomplished through carbonate weathering via either the sulphuric acid derived from pyrite oxidation or carbonic acid. Calcium can also be sourced as a product of silicate weathering. Whilst silicate-derived potassium does vary with sulphate concentrations, ranging from c.20-300μmol/l over the measured range of $SO_4^{2-}$, the lower proportion of other purely silicate derived cations and silicic acid in

landslide seepage (e.g. $Na^+$:$Ca^{2+}$ ratio always <0.5, compared to the silicate mineral value of 2.85 (Calmels et al. 2011)), as well as the markedly different behaviour of sodium and calcium (Figure 3A and 3B) supports the interpretation that weathering in the sampled landslides is dominated by the dissolution of carbonates.

An implication of our observations is that in the Taimali catchment, a significant proportion of the sedimentary pyrite contained within the rock mass survives exhumation, and is only exposed by landslide fragmentation at the surface. Previous studies in Taiwan (Calmels et al. 2011) have shown that Sulphate is a key component in deep groundwater (sampled >100m below the surface), but clearly not all pyrite is oxidised at these depths. Groundwater circulating in the bedrock topography is unlikely to be the source of sulphate in seepage. Such water would not systematically surface at landslide sites, given the highly variable morphologic characteristics (size, scar/deposit ratio, position with respect to ridge crest and valley floor, and proximity to major faults) of individual landslides. The same argument applies to geothermal outflow. Moreover, hot spring waters sampled in the Taimali catchment have extremely high Lithium concentrations, 1610 μmol/l, and the lack of elevated $Li^+$ concentrations in the landslide seepage samples suggests the elevated sulphate in seepage derives from in-situ weathering. Other studies have found high concentrations of silicate-derived cations in deeper groundwater (Calmels et al. 2011); we do not observe any strong increase in the proportion of cations exclusively sourced from silicate minerals (particularly sodium – figure 3B) and sulphate in landslide seepage.

Extremely elevated $Na^+$ and $Mg^{2+}$ concentrations at some sites are not clearly associated with any other measured parameter, or with each other, and we have no ready explanation for these isolated observations at present. Since these high concentrations are not linked to dissolved sulphate, they may be linked to flowpaths with limited oxygen where sulphate oxidation cannot proceed, but we do not have a constraint on this.

Overall, we suggest that the spatial heterogeneity of groundwater flow paths in mountain belts (Andermann et al. 2012, Calmels et al. 2011) and extensive tectonic fracturing (Molnar et al. 2007, Clarke and Burbank 2011; Menzies et al. 2014) will to some extent preclude the existence of well-defined depletion depths for reactive mineral phases as seen elsewhere (e.g. Brantley et al. 2013, Drake et al. 2009). As such, some reactive phases likely reach the near-surface where landslides can expose them to oxygenated fluid.

### 5.2. Landslide weathering products in stream water

To directly compare the impact of landsliding on weathering in the steep mountains of Taiwan between landscape units with variable landslide activity, we have used the volumes of landslides calculated for each sampled subcatchment within the Taimali drainage area, normalised to catchment area and evaluated this against the solute load of the streams. For this purpose, we use landslide volume rather than surface area, because much of the weatherable mineral surface area is contained within the landslide debris, rather than at the landslide surface. In doing so, we assume that the hydraulic connectivity in the groundmass of the landslide is always high enough to wet the mineral surface area within the entire landslide body. We find that TDS and $Ca^{2+}$ increase with increased landslide volumes in the sampled subcatchments where the landslide volume density exceeds approximately $10^4 m^3/km^2$ of catchment area (Fig. 4). Above this 'threshold', we observe a doubling of TDS from c.6000 to more than 12000 μmol/l and increases in the concentration of $Ca^{2+}$ from c.1000 to nearly 5000 μmol/l, as landslide densities increase up to c.$10^6$ $m^3/km^2$. There is an increase in the $SO_4^{2-}$ over this same landslide density range, from c.300 to nearly 4000μmol/l, and as in the individual landslide seepage samples, there is a strong correlation of both stream TDS and dissolved $Ca^{2+}$ concentrations with the dissolved Sulphate ($R^2$ = 0.929 and 0.9318, respectively; p<0.001 for both relationships). Dissolved $Na^+$ (Fig.5A) does

not systematically vary with $SO_4^{2-}$ (p = 0.74), although $K^+$ does increase significantly, from 15-130μmol/l with Sulphate concentration (p <0.001). At lower specific landslide volumes, there is a larger degree of variability and below $10^4 m^3/km^2$, equivalent to a debris layer with a uniform thickness of about 1 cm over the catchment, the stream TDS and $Ca^{2+}$ concentrations do not vary systematically with the landslide volume.

The importance of landsliding in setting the chemical load of streams varies depending on which element is considered.
For example, landslides exert a strong control over dissolved Calcium and Potassium in streams, but have almost no impact on dissolved Sodium. At the catchment scale, solute concentrations remain primarily dependent on the additional acidity provided by oxidation of pyrite; coupled carbonate-sulphuric acid weathering increases the $Ca^{2+}$ output, and with it of the total exported solute load. On the other hand, dissolved $SiO_2$ is unaffected by either increasing landslide volume or changes in $SO_4^{2-}$. Notably, values of $SiO_2$ measured in the streams of the Taimali catchment can exceed those
measured in landslide seepage, suggesting that any landslide-induced impact on local silicate weathering is not significant enough to outweigh weathering elsewhere, either in soils on stable hillslopes or deeper in the subsurface. Longer flow paths of deep groundwater, reaching well below the levels affected by landsliding, are more likely to be enriched in slower dissolving silicate weathering products (Bickle et al. 2015; Calmels et al. 2011; Galy and France-Lanord 1999), and the relatively low fraction of sodium and silicic acid in landslide seepage suggests a comparatively
short water residence time in mass wasting deposits (Maher 2011). In view of this, we posit that the disconnection between stream-borne Sulphate and some silicate-weathering products ($Na^+$, silicic acid), juxtaposed with a link of Sulphate and Potassium, which is also only sourced from silicate, reflects the existence of multiple pathways relevant for silicate weathering on a catchment scale. More congruent weathering in deep groundwater likely complements the rapid weathering of carbonates and incongruent weathering of silicates (such as rapid K release from chlorite and
biotite, (Malmstrom et al. 1996)) in mass wasting deposits.

The sampled subcatchments have up to 16.6% of the total area occupied by landslide deposits and scars. However, the Taimali watershed contains some exceptionally large landslides, the outlines of which define tributary catchments in their own right. The largest landslide we sampled (TWS15-46) has a surface area of $2.7km^2$. At these very large scales and where landsliding has affected such an important part of the topography, the distinction between landslide and
stream catchment becomes blurred, and solute concentrations from such catchments will also be subject to the same limits on TDS concentrations in the landslides themselves. The peak TDS values we measured in seepage are between 15000-20000μmol/l. Other locations may have greater concentrations, if local weathering has greater oxidation of sulphides, either due to more efficient distribution oxygenated water at depth in the landslide deposit. Such hypothetical, very high concentrations, in excess of what we have recorded, would serve as an upper limit to solute
concentrations in the entire catchment.

Even where landsliding is less prominent, there is still significant variability in tributary stream dissolved load (Fig.4). This is unlikely to be an artefact of the mapping; landslides below the detection limit (about $500 m^2$) are unlikely to mine significant quantities of bedrock. Instead, they primarily mobilize colluvial or soil material, which limits their impact via the oxidation of sulphides. Meanwhile, other environmental factors that result in variability in TDS at higher
landslide rates will also apply to the steep mountain landscape of southern Taiwan in the absence of landslides. This includes variable oxidation of pyrite in a tectonically fractured substrate, which will affect both deep and shallow groundwater fluxes, depending on where in the weathering zone the respective hydrological flow paths locate. Other studies have incorporated topographic factors, such as local slope or hillslope length, into the models of weathering (Heimsath et al. 1997; Maher and Chamberlain 2014). These factors may also be a source of variability in

subcatchments with low landslide rates. However, landsliding during typhoon Morakot was primarily controlled by the intensity and volume of rainfall (Lin et al. 2011), and the pattern of landsliding in the Taimali catchment is not strongly tied to topographic characteristics. Hence, the amount of mass wasting in individual sub-catchments is not strongly related to the local modal slope (see data in Table 2). Moreover, neither modal slope nor the range of elevation in a catchment is correlated with the measured TDS, suggesting these factors do not strongly control the weathering in this
setting.

Many subcatchments of the Taimali watershed contain significant amounts of alluvial material, which derives from landslide debris delivered to streams. Alluviated streambeds could be a locus of enhanced weathering, but we do not observe any correlation between the measured stream chemistry and the proportion of the catchment filled with alluvial material in the six subcatchments with significant fill (>1% of total catchment area – see Table 1. $R^2$ = -0.365; Kendall's
Tau = -0.067; p = 0.48). However, due to the lack of rainfall prior to and during the sampling period we were unable to sample hyporheic water from within the alluvial streambeds. Presence of secondary precipitates in fill deposits suggests that at low flow these sedimentary bodies could act as sinks for solutes, rather than sources. Although the role of alluvial materials in the weathering budget of the Taimali catchment is not fully clear, our observations suggest that it is not a first order control on the chemical output during the sampling interval.
As we did not measure the discharge in these small streams we can only report concentrations of solutes. Weather-determined dilution could affect the measured solute concentrations and cause them to vary between catchments. However, observations from other parts of the Taiwanese mountains (Calmels et al. 2011, Lee et al. 2015) suggest that dissolved Calcium concentrations are only diluted by more than approximately 35% during peak typhoon discharge conditions in mountain rivers. Therefore, we expect that although the period when we sampled was relatively dry, our
measurements are a fair first order representation of solute concentrations and ratios during the majority of the year. This is consistent with observations from other mountain belts, including the Southern Alps of New Zealand, the Himalayas and the Andes, where rivers exhibit near-chemostatic behaviour over a range of discharge conditions (Lyons et al. 2005; West et al. 2005, Torres et al. 2015, Tipper et al. 2006).

**5.3. Landsliding and the weathering of labile minerals**

Solute concentrations in landslide seepage and stream water in the Taimali catchment are significantly higher than in the rivers of the WSA by approximately 5-10 times (Fig.4a). Both locations share a steep, landslide-dominated mountain physiognomy and very high rainfall rates, but Taiwan is warmer and has carbonate rich, pyrite bearing lithologies, whereas the WSA formations have only trace amounts of these minerals. We suggest that lithology and climate set a
baseline for weathering in a given setting, above which landslides can control the weathering variability. This control is especially clearly expressed in the WSA, where the substrate for weathering is relatively homogeneous. The Taiwanese river chemistry has a larger spread with respect to landslide volumes, but the clear correlation between stream TDS and $SO_4^{2-}$ (Fig.5A, $R^2$ = 0.93, p <0.001) reflects the importance of the weathering occurring within the landslides in setting catchment scale river solute concentrations. It is unlikely that this correlation arises from dilution or mixing of sources.
The lack of correlation between sodium and sulphate concentrations (Fig 5B, p = 0.74) rules out a strong influence of both dilution and evapotransporation, and likely precludes weathering pathways with elevated sulphate and silicate weathering such as deeper groundwater from contributing to the higher TDS. Heterogeneous distribution of sulphides in the bedrock substrate of the Taimali catchment and its variable oxidation during exhumation likely not only affect the individual landslide seepage concentrations, but also the chemistry of streams throughout the catchment. This may

explain why the total volume of landslides is a weaker control over stream TDS than in New Zealand.

Reactions of the most labile mineral phases dominate the chemistry of landslide seepage. In the Taimali catchment and elsewhere in Taiwan, the fastest reaction is the oxidation of sulphide. In the case of the WSA, elevated landslide seepage concentrations (Emberson et al. 2015) result from weathering of small amounts of highly labile carbonate,

which become exposed through rock mass fragmentation. However, unlike the Taimali, the fragmentation of WSA bedrock does not introduce an extra source of acidity. The relative abundance of carbonate in Taiwan means fluids are supersaturated everywhere with respect to calcite (Table 1). Therefore, the potential for landsliding to increase carbonate weathering due to rock mass fragmentation would be limited but for the exposure of pyrite and the associated generation of sulphuric acid. Thus, exposure of reactive minerals due to rock mass fragmentation boosts weathering

rates in landslides, but the minerals and their roles vary between settings. The concept of a reaction front for weathering is less applicable to these settings, as the distribution of reactions will be controlled by the localisation of detrital sulphides and the flow paths providing oxygenated fluid.

Landslide weathering rates will remain elevated until the relevant reactive mineral(s) have been depleted, after which weathering in landslides is likely to proceed much like in other parts of the landscape, driven by organic and carbonic

acids. The initial abundance of trace sulphide and/or carbonate together with their exhaustion, over longer time scales, is likely to be an important control on the duration of rapid weathering in landslides. However, in choosing the Taimali catchment, where the vast majority of the mass wasting observed occurred very recently and all at once, we have also selected a location where the long-term impact of rock mass fragmentation in landslides is difficult to observe. Nevertheless, some constraints may be gleaned from two subcatchments (TWS15-40 and TWS15-53) where the TDS

and dissolved $SO_4^{2-}$ values are lower than others with similarly high landslide incidence (TWS15-40 – 9220µmol/l; TWS15-53 – 7617µmol/l compared to other streams with TDS greater than 12000µmol/l). These two subcatchments have large landslides that had been moving persistently for at least two decades prior to typhoon Morakot. Although these landslides were reactivated during the typhoon, they likely displaced materials that had already been significantly depleted of pyrite due to prior fragmentation and weathering. This is circumstantial evidence of a substantial loss of

weathering intensity by depletion of especially labile minority minerals on a decadal time scale in this instance, similar to previous findings in the WSA (Emberson et al. 2015).

A second time scale of relevance is that of the physical removal of landslide debris from the catchment. If debris remains in a catchment after depletion of exposed pyrite, then its weathering will revert to mineral reactions with Carbonic Acid. If the time to depletion is greater than the time required for removal of the debris, then the labile phases

it contains will remain unweathered and will be removed through sediment transport processes. In the case of Taiwan, some samples of river bed material show Sulphur content of up to 1.07% (Hilton et al. 2014), suggesting that weathering in the landscape does not purge all sulphides brought to the surface by rock mass exhumation. In the opposite case the weathering from a given landslide will return to a constant background rate shared with other colluvium and soil covered parts of the landscape. How this scales up to the catchment scale depends on the return time

of large drivers of mass-wasting, such as typhoons and earthquakes, and the subsequent advection of debris into the fluvial network and the transport capacity of this network. A disparity between short term production of sediment on hillslopes and the capacity for onward transport can lead to up to millenial residence times of event-produced sediment in mountain landscapes (Blöthe and Korup 2013). In the Taimali catchment, many headwater streams were still clogged with sediment 6 years after typhoon Morakot, and large colluvial fans had formed below several landslide-dominated

tributaries. Elsewhere in Taiwan, mountain valleys contain large colluvial terraces, which have formed hundreds to thousands of years ago (Hsieh and Chyi 2010). This implies that time scales of pyrite depletion and of debris removal must both be considered in further explorations of the links between erosion and weathering in Taiwan.

## 5.4. Impact on climate

The link between weathering and atmospheric $pCO_2$ is determined by the mineralogy of the weathered substrate, the composition of the weathering acid, and the residence time of material in the weathering zone (Torres et al. 2016, Galy and France-Lanord 1999, Lerman et al. 2007). Therefore, the balance of silicate:carbonate weathering, the source of acidity in mass-wasting deposits and the retention of these sediments in the landscape are all crucial to the role of landsliding in the Carbon cycle. As a result of this complexity, the impact of landsliding on climate is ambiguous.

Weathering of silicates with Carbonic acid can lower the atmospheric $pCO_2$, but with Sulphuric acid it has no effect. Dissolution of carbonates has no net effect on atmospheric $pCO_2$, if achieved with Carbonic acid, and it can increase $pCO_2$ if it is accomplished with Sulphuric acid (Calmels et al. 2007; Gaillardet et al. 1999; Torres, West, and Li 2014). As erosion rates increase and mass-wasting through bedrock landsliding begins to dominate erosional budgets, the erosional impact on $CO_2$ and climate, achieved through sustained weathering of silicates (Berner and Kothavala 2001;

Brady 1991, Raymo and Ruddiman 1992), will weaken. Weathering of carbonates will form a greater portion of solute budgets, even where they are only present in small amounts, and in mountain belts where pyrite is ubiquitous, erosion-driven weathering may even become a source of $CO_2$ (e.g. Calmels et al. 2007, Torres et al. 2016).

The length of time over which sediment is stored in a catchment will determine the cumulative effect of this weathering. In Taiwan, storage times of greater than 50kyrs have been demonstrated for sedimentary terraces derived from mass-

wasting (Hsieh et al. 2013, 2010). In such locations, where the retention time of sediment derived from mass-wasting is long compared to the decay time of the weathering boost provided by the highly labile minerals contained initially in these sediments, the coupled sulphuric acid-carbonate weathering will likely be superseded by slower silicate weathering, reducing any net release of $CO_2$. When the sediment retention time equals the weathering boost decay time, the effect of landsliding on $CO_2$ release is maximized. Shorter sediment residence times promote the export of

unweathered sediment to the ocean and its burial in marine basins. This can result in a fraction of sulphides contained in the rock mass bypassing the weathering window (Hilton et al. 2014), reducing the potential effect of their exhumation on atmospheric $pCO_2$ and returning the landscape to a state in which distributed weathering in soils and colluvial fills dominates. In addition to this inorganic complexity, the export and burial of organic carbon via landsliding also has an impact on the Carbon budget of Taiwan (Hilton et al. 2008, 2011; West et al. 2011). Ultimately, quantification of the

processes side by side is important to understanding of the role of mountain erosion in the Earth's Carbon cycle.

In the Taimali catchment – a good analogue for many young mountain belts formed of continental margin sediments – widespread, typhoon-triggered landsliding has caused a significant increase in carbonate weathering by Sulphuric acid over timescales shorter than the return time of extreme meteorological drivers of mass wasting. As a result, the area

most affected by erosion due typhoon Morakot could be a net source of $CO_2$ to the atmosphere. The weathering boost is likely to end before the next major meteorological perturbation of southern Taiwan, so that the weathering of carbonates and silicates with Carbonic acid must be taken into account when considering the longer-term effects of erosion in South Taiwan. Future work to quantify the net sequestration or release of $CO_2$ would require a closer evaluation of the ratios of acid at work, as well as correction for the extensive secondary precipitation evident in the catchment.

## 6. Conclusions

Landslides can mobilize and fragment large volumes of rock mass from below the soil and regolith mantle of steep hillslopes. Weathering of this material is promoted by the exposure of fresh rock to infiltrating water and oxygen in landslide debris with very high internal surface area. Under these conditions, weathering is limited by the kinetics of the

reactions of the most labile mineral phases, but as long as these phases are abundant in the rockmass the rate of weathering is vastly increased. We have found that these labile phases may be minority constituents, driving the weathering in fresh landslide deposits away from the slower process affecting the surface materials elsewhere in the landscape over longer time scales.

In Southern Taiwan, weathering in landslides triggered during typhoon Morakot in 2009, is governed by the rapid

oxidation of pyrite, which is a trace mineral in the pelitic rocks of the local substrate. This oxidation generates sulphuric acid, which reacts primarily with ubiquitous carbonates. Five years after landsliding, carbonic acid is a weathering agent of lesser importance, while the weathering of silicate minerals is not elevated. In the Taimali catchment in Southeast Taiwan, which sustained landsliding at spatially diverse rates during typhoon Morakot, stream chemistry suggests that landslides have tapped into rock mass with variable amounts of pyrite. Although total dissolved solids concentrations in

tributary streams vary with the landslide density in their subcatchments, they are better correlated with the Sulphate concentration. In rare subcatchments where older landslides, dating back at least 25 years, dominate, TDS and Sulphate concentrations are markedly lower, indicating that the effects of exposure and oxidation of pyrite in landslides may dissipate on decadal time scales in this area.

Southern Taiwan sits within a rapidly deforming and uplifting mountain belt, formed with calcareous sandstones and

pelites from the Asian continental margin. It shares its deformation style and a range of lithologies with many other young mountain belts at convergent plate boundaries. Bedrock landsliding is the dominant mode of mass wasting and sediment production in such settings. Therefore, we submit that our observations of landslide weathering in Taiwan may well have wider implications. Moreover, even where pyrite is not significantly present, the importance of labile mineral phases in weathering in recent landslide deposits is evident. In the Western Southern Alps of New Zealand, the rapid

reaction of carbonates, which make up only a very small fraction of the rock mass, with carbonic acid dominates the chemistry of landslide seepage and helps set the weathering flux from the mountain belt (e.g. Jacobson et al. 2003, Emberson et al. 2015). The combination of results from these two settings suggests that landslides may affect weathering in other rapidly denuding mountain belts, albeit modulated by lithology.

Hence, in order to understand chemical weathering rates from mountain belts where erosion is dominated by stochastic

mass-wasting it is crucial to consider the way in which this specific erosive process controls weathering. We propose a conceptual model of a rapidly eroding landscape where the background weathering is set by lithology and climate, overprinted by the impact of landslides. The importance of reactive mineral phases, weathering rapidly in small and specific parts of the landscape affected by deep-seated erosion, is a contrast to existing concepts, which do not encompass the stochastic nature of weathering (Dixon and von Blanckenburg 2012; Maher and Chamberlain 2014;

Riebe et al. 2003)A full description of the impact of landsliding on weathering should include the effect on the availability of reactive phases in addition to the previously modelled spatial stochasticity (Gabet 2007).

Ultimately, the importance and impact of landslide weathering will be determined by the rate of the geomorphic process and the mineral composition of the substrate in which it occurs. However, it will also depend on the efficiency with which the products of mass wasting are removed from the landscape and transferred into geological basins, relative to

the time needed to leach the most labile mineral phases from the mobile sediment, and to the return time of important mass wasting events in the landscape. In this light, we draw attention to the possibility that disproportionate dissolution of highly labile mineral phases such as pyrite or carbonate as a result of landslide driven weathering may reduce the strength of the link between erosion and atmospheric $CO_2$ drawdown via silicate weathering in fast eroding settings, which in turn would modulate any feedback on erosion through changes in tropical storm intensity (e.g. Knutson et al.

2010). The role of active mountain belts in the global Carbon cycle remains ambiguous.

## 7. Acknowledgements

We thank Meng-Long Hsieh and several of his students for sampling support and local knowledge in the Taiwan. Assistance from Barney Ward, Claire Nichols and Kristen Cook greatly aided field campaigns. Carolin Zorn measured

the anion content of the samples, and Rene Mania significantly improved landslide maps for Taiwan covering the last 20 years. Discussion with Friedhelm von Blanckenburg and Josh West aided the development of the paper. R.E., N.H., and A.G conceived the study; R.E., N.H., and O.M. collected samples; O.M. generated landslide data; R.E. wrote the paper with major input from all authors. Three anonymous reviewers greatly aided the development of the manuscript with constructive commentary. R.E. has benefited from Helmholtz Institutional funding during this project.

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

**Figure Captions**

**Figure 1**

Locations of sampling. Fig.1A: location of Taimali catchment and landslides sampled outside of this river in Taiwan.

Fig.1B: Taimali river, with individual streams outlined over a relief map derived from ASTER DEM. Streams are colour

coded by the fraction of their area occupied by landslides (scars and deposits); individual landslides are also highlighted

as they were mapped from satellite imagery, together with the alluvial fill in the channel. Locations where we sampled

landslide seepage are also shown.

**Figure 2**

Total dissolved solids (TDS) in landslide seepage plotted against the mapped area of the individual landslides.

Analytical error on TDS measurements are shown.

**Figure 3**

Above: molar ratio of dissolved calcium to sum of dissolved calcium, magnesium, potassium and sodium ($Ca^{2+}/Na^{+}+K^{+}+Ca^{2+}+Mg^{2+}$) plotted against ratio of dissolved sulphate to the same dissolved elements ($SO_4^{2-}/Na^{+}+K^{+}+Ca^{2+}+Mg^{2+}$).

Below: molar ratio of dissolved sodium to sum of dissolved calcium, magnesium, potassium and sodium ($Na^{+}/Na^{+}+K^{+}+Ca^{2+}+Mg^{2+}$) plotted against ratio of dissolved sulphate to the same dissolved elements ($SO_4^{2-}/Na^{+}+K^{+}+Ca^{2+}+Mg^{2+}$).

Error bars are the propogated total measurement uncertainty.

**Figure 4**

Normalised volume of landslides (calculated volume divided by catchment area) plotted against total dissolved solids

(TDS, fig A), dissolved Calcium ($Ca^{2+}$, fig B), and dissolved Sulphate ($SO_4^{2-}$, fig C) for streams in the Taimali river (this

study) and previous numbers for catchments in the Western Southern Alps (WSA) of New Zealand (Emberson et al.

2016). Vertical error bars are total measurement uncertainty; horizontal error bars are 95% confidence intervals for

volumes. Each chemical measurement has been corrected for cyclic and hydrothermal input.

**Figure 5**

(A) Dissolved Sulphate ($SO_4^{2-}$) plotted against total dissolved solids (TDS, above) and (B) dissolved sodium ($Na^+$, below) in streams in the Taimali river. Error bars are total measurement uncertainty.

Table 1.

| Name | Latitude | Longitude | pH | T, °C | $Ca^{2+}$, μmol/l (±3%) | $K^+$, μmol/l (±5%) | $Li^+$, μmol/l (±2%) | $Mg^{2+}$, μmol/l (±4%) | $Na^+$, μmol/l (±6%) | Si, μmol/l (±3%) | $Sr^{2+}$, μmol/l (±2%) | $Cl^-$, μmol/l (±10%) | $SO_4^{2-}$, μmol/l (±10%) | $HCO_3^-$, μmol/l (±10%) | TDS, μmol/l | Landslide area, m² (±10%) | Landslide volume, m³ | Calcite saturation index |
|---|---|---|---|---|---|---|---|---|---|---|---|---|---|---|---|---|---|---|
| **Taimali & Surrounds Seepage** | | | | | | | | | | | | | | | | | | |
| TWS15-17 | 22.52297 | 120.90475 | 7.63 | 22.2 | 2716 | 73 | 1.92 | 1016 | 571 | 365 | 17 | 128 | 1453 | 5109 | 11450 | 83825 | 173825 | 0.3 |
| TWS15-26 | 22.59528 | 120.94849 | 8.23 | 21.0 | 2862 | 43 | bdl | 1117 | 590 | 201 | 18 | 127 | 2322 | 3851 | 11131 | 33676 | 51591 | 0.7 |
| TWS15-31 | 22.58965 | 120.92747 | 8.06 | 21.6 | 1534 | 25 | bdl | 857 | 469 | 232 | 13 | 129 | 789 | 3595 | 7643 | 7589 | 7089 | 0.4 |
| TWS15-33 | 22.58543 | 120.92238 | 8.45 | 21.3 | 1324 | 42 | bdl | 925 | 523 | 289 | 14 | 142 | 798 | 3353 | 7410 | 40666 | 66325 | 0.8 |
| TWS15-38 | 22.57942 | 120.89299 | 8.68 | 21.1 | 1019 | 48 | 5.82 | 3679 | 559 | 195 | 3 | 101 | 1965 | 5979 | 13555 | 14156 | 16264 | 0.9 |
| TWS15-41 | 22.57541 | 120.87698 | 7.67 | 21.7 | 3374 | 43 | bdl | 185 | 334 | 367 | 22 | 42 | 1198 | 5100 | 10665 | 35180 | 54682 | 0.4 |
| TWS15-42 | 22.56998 | 120.87584 | 8.36 | 19.5 | 2123 | 21 | bdl | 373 | 317 | 226 | 18 | 53 | 1366 | 2582 | 7080 | 16968 | 20704 | 0.7 |
| TWS15-46 | 22.57503 | 120.82355 | 7.30 | 21.4 | 4462 | 168 | bdl | 179 | 365 | 282 | 13 | 18 | 2851 | 4185 | 12523 | 2775932 | 63566055 | -0.2 |
| TWS15-47 | 22.58521 | 120.8213 | 8.26 | 21.5 | 3946 | 41 | bdl | 240 | 415 | 289 | 14 | 28 | 2714 | 3400 | 11088 | 95319 | 206275 | 0.7 |
| TWS15-49 | 22.57556 | 120.8312 | 8.29 | 21.0 | 2588 | 46 | bdl | 41 | 271 | 257 | 11 | 38 | 1503 | 2552 | 7306 | 123628 | 291661 | 0.6 |
| TWS15-51 | 22.57318 | 120.85339 | 8.26 | 24.5 | 2975 | 60 | bdl | 545 | 381 | 267 | 22 | 40 | 2450 | 2583 | 9322 | 19656 | 25183 | 0.6 |
| TWS15-63 | 22.51877 | 120.68069 | 7.49 | 22.4 | 1475 | 34 | 3.42 | 1267 | 796 | 162 | 6 | 27 | 661 | 4965 | 9397 | 50139 | 87662 | 0.1 |
| TWS15-65 | 22.53412 | 120.69309 | 8.39 | 27.8 | 1666 | 77 | 7.52 | 3476 | 452 | 173 | 5 | 21 | 1867 | 7069 | 14813 | 214670 | 608272 | 1.0 |
| TWS15-68 | 22.76694 | 120.66478 | 7.99 | 23.7 | 4337 | 29 | 4.38 | 1643 | 406 | 199 | 9 | 26 | 4080 | 4226 | 14960 | 15609 | 18525 | 1.1 |
| TWS15-72 | 22.73699 | 120.71692 | 7.83 | 20.8 | 1680 | 27 | 4.80 | 1081 | 807 | 154 | 8 | 17 | 1336 | 3674 | 8788 | 132000 | 318261 | 0.3 |
| TWS15-73 | 22.73627 | 120.71718 | 7.64 | 19.5 | 1926 | 33 | 6.19 | 1412 | 963 | 131 | 9 | 16 | 1891 | 3879 | 10267 | 61035 | 113912 | 0.1 |
| TWS15-86 | 22.44895 | 120.88542 | 8.46 | 20.5 | 1373 | 26 | 3.45 | 1187 | 541 | 244 | 6 | 175 | 749 | 4026 | 8332 | 30188 | 44598 | 1.0 |
| TWS15-87 | 22.58739 | 120.8484 | 8.43 | 22.4 | 3308 | 39 | bdl | 229 | 312 | 205 | 19 | 34 | 2344 | 2739 | 9229 | 17940 | 22298 | 1.2 |
| TWS15-90 | 22.60323 | 120.84421 | 7.76 | 29.8 | 4658 | 112 | bdl | 153 | 305 | 230 | 16 | 28 | 3118 | 3810 | 12431 | 23787 | 32469 | 0.9 |
| **Chenyoulan seepage** | | | | | | | | | | | | | | | | | | |
| CH1307-101 | 23.61785 | 120.89787 | 7.98 | 32.4 | 8344 | 85 | 9.94 | 1189 | 983 | 287 | 36 | 30 | 7827 | 1189 | 23292 | 121000 | 284460 | 1.3 |
| CH1310-40 | 23.62402 | 120.89918 | 7.46 | 23.4 | 3813 | 42 | 5.71 | 1448 | 638 | 274 | 26 | 27 | 3649 | 1448 | 13857 | 13528 | 15310 | 0.4 |
| **Taimali Streams** | | | | | | | | | | | | | | | | | | |
| TWS15-23 | 22.59175 | 120.93216 | 8.42 | 20.5 | 1164 | 21 | bdl | 478 | 381 | 225 | 7 | 75 | 454 | 2699 | 5505 | | | 0.8 |
| TWS15-25 | 22.59484 | 120.94132 | 8.40 | 20.2 | 1556 | 25 | bdl | 670 | 515 | 272 | 11 | 101 | 969 | 2976 | 7094 | | | 0.9 |
| TWS15-32 | 22.58802 | 120.92323 | 8.34 | 20.1 | 1801 | 0 | bdl | 876 | 513 | 269 | 13 | 128 | 1296 | 3173 | 8069 | | | 0.9 |
| TWS15-34 | 22.58496 | 120.92101 | 8.42 | 19.7 | 892 | 19 | bdl | 237 | 377 | 287 | 6 | 100 | 235 | 2077 | 4229 | | | 0.6 |
| TWS15-35 | 22.58763 | 120.91415 | 8.51 | 20.6 | 1535 | 27 | bdl | 773 | 326 | 190 | 11 | 55 | 1014 | 2901 | 6830 | | | 1.0 |
| TWS15-36 | 22.58109 | 120.91301 | 8.56 | 19.8 | 1357 | 20 | bdl | 562 | 381 | 201 | 8 | 99 | 533 | 3082 | 6243 | | | 1.0 |
| TWS15-39 | 22.57508 | 120.88979 | 8.47 | 20.5 | 1309 | 17 | bdl | 614 | 318 | 206 | 5 | 59 | 660 | 2799 | 5988 | | | 0.9 |
| TWS15-40 | 22.57427 | 120.88347 | 8.40 | 19.9 | 2627 | 76 | bdl | 974 | 322 | 186 | 30 | 35 | 2622 | 2383 | 9256 | | | 0.9 |
| TWS15-43 | 22.56971 | 120.87585 | 8.47 | 18.3 | 1326 | 30 | bdl | 350 | 330 | 219 | 9 | 61 | 617 | 2435 | 5377 | | | 0.8 |

| Sample Name | | | | | | | | | | | | | | | |
|---|---|---|---|---|---|---|---|---|---|---|---|---|---|---|---|
| TWS15-44 | 22.57248 | 120.82725 | 8.53 | 18.5 | 1732 | 18 | bdl | 64 | 247 | 239 | 6 | 30 | 846 | 2145 | 5327 | 0.9 |
| TWS15-48 | 22.58458 | 120.81979 | 7.77 | 20.0 | 2703 | 30 | bdl | 106 | 169 | 154 | 7 | 15 | 1487 | 2850 | 7521 | 0.5 |
| TWS15-50 | 22.56602 | 120.84565 | 8.46 | 22.1 | 2069 | 34 | bdl | 137 | 301 | 268 | 8 | 40 | 1778 | 1167 | 5802 | 0.7 |
| TWS15-52 | 22.57761 | 120.85395 | 8.19 | 27.1 | 2736 | 78 | bdl | 138 | 305 | 225 | 13 | 26 | 2280 | 1574 | 7374 | 0.7 |
| TWS15-53 | 22.57609 | 120.86786 | 8.46 | 25.5 | 1678 | 61 | bdl | 1281 | 296 | 161 | 14 | 37 | 2138 | 1990 | 7657 | 0.8 |
| TWS15-88 | 22.58842 | 120.84806 | 8.25 | 22.5 | 4642 | 78 | bdl | 346 | 480 | 296 | 21 | 42 | 3734 | 3068 | 12709 | 1.1 |
| TWS15-89 | 22.59241 | 120.84619 | 8.46 | 22.0 | 1902 | 40 | bdl | 146 | 348 | 275 | 9 | 35 | 948 | 2571 | 6275 | 1.0 |
| TWS15-93 | 22.62438 | 120.8435 | 8.27 | 25.1 | 4815 | 129 | bdl | 172 | 273 | 217 | 22 | 20 | 3761 | 2880 | 12287 | 1.2 |
| TWS15-94 | 22.59696 | 120.84148 | 8.41 | 24.3 | 2962 | 49 | bdl | 134 | 336 | 226 | 12 | 26 | 1933 | 2711 | 8388 | 1.1 |
| Hot Spring | | | | | | | | | | | | | | | | |
| TWS15-22 | 22.59343 | 120.93897 | 8.43 | 24.1 | 71 | 862 | 1611 | 28 | 46541 | 3643 | 8 | 40124 | 235 | 7011 | 100150 | -0.2 |


## Table 2: Taimali stream characteristics

| Sample Name | Latitude | Longitude | Modal Slope | Elevation range, m | Mean Elevation, m above s.l. | Fractional Catchment landslide area | Fractional Catchment landslide volume | Catchment area, m² | Fractional Channel Fill pre Morakot | Fractional Channel Fill Post Morakot |
|---|---|---|---|---|---|---|---|---|---|---|
| Taimali Streams | | | | | | | | | | |
| TWS15-23 | 22.59175 | 120.93216 | 26 | 1030 | 734 | 0.0002 | 0.0001 | 3129659 | | |
| TWS15-25 | 22.59484 | 120.94132 | 30 | 661 | 508 | | | 496256 | | |
| TWS15-32 | 22.58802 | 120.92323 | 32 | 285 | 364 | | | 8467 | | |
| TWS15-34 | 22.58496 | 120.92101 | 22 | 732 | 547 | 0.0089 | 0.0203 | 1023265 | | |
| TWS15-35 | 22.58763 | 120.91415 | 24 | 1196 | 790 | 0.0010 | 0.0014 | 9861469 | | |
| TWS15-36 | 22.58109 | 120.91301 | 28 | 799 | 643 | 0.0030 | 0.0077 | 3359874 | | |
| TWS15-39 | 22.57508 | 120.88979 | 23 | 872 | 723 | 0.0350 | 0.0414 | 3652240 | | |
| TWS15-40 | 22.57427 | 120.88347 | 28 | 1126 | 740 | 0.1665 | 0.7246 | 5461061 | 0.0325 | 0.0328 |
| TWS15-43 | 22.56971 | 120.87585 | 33 | 781 | 711 | 0.0180 | 0.0857 | 1211594 | | |
| TWS15-44 | 22.57248 | 120.82725 | 40 | 745 | 868 | 0.0740 | 0.1187 | 357320 | | |
| TWS15-48 | 22.58458 | 120.81979 | 33 | 2653 | 1473 | 0.1170 | 0.2774 | 62281990 | 0.0218 | 0.032 |
| TWS15-50 | 22.56602 | 120.84565 | 23 | 1007 | 898 | 0.0690 | 0.1005 | 4423928 | | |
| TWS15-52 | 22.57761 | 120.85395 | 31 | 1308 | 795 | 0.0880 | 0.2111 | 25323473 | 0.0466 | 0.1268 |
| TWS15-53 | 22.57609 | 120.86786 | 26 | 959 | 805 | 0.1480 | 1.2017 | 3012380 | 0.0214 | 0.0332 |
| TWS15-88 | 22.58842 | 120.84806 | 31 | 225 | 631 | 0.1293 | 0.1646 | 145453 | | |
| TWS15-89 | 22.59241 | 120.84619 | 29 | 484 | 666 | 0.0170 | 0.0637 | 596003 | | |
| TWS15-93 | 22.62438 | 120.8435 | 33 | 977 | 1115 | 0.1660 | 0.3246 | 5670550 | 0.0113 | 0.0348 |
| TWS15-94 | 22.59696 | 120.84148 | 26 | 1032 | 806 | 0.1150 | 0.1833 | 4462662 | 0 | 0.0512 |

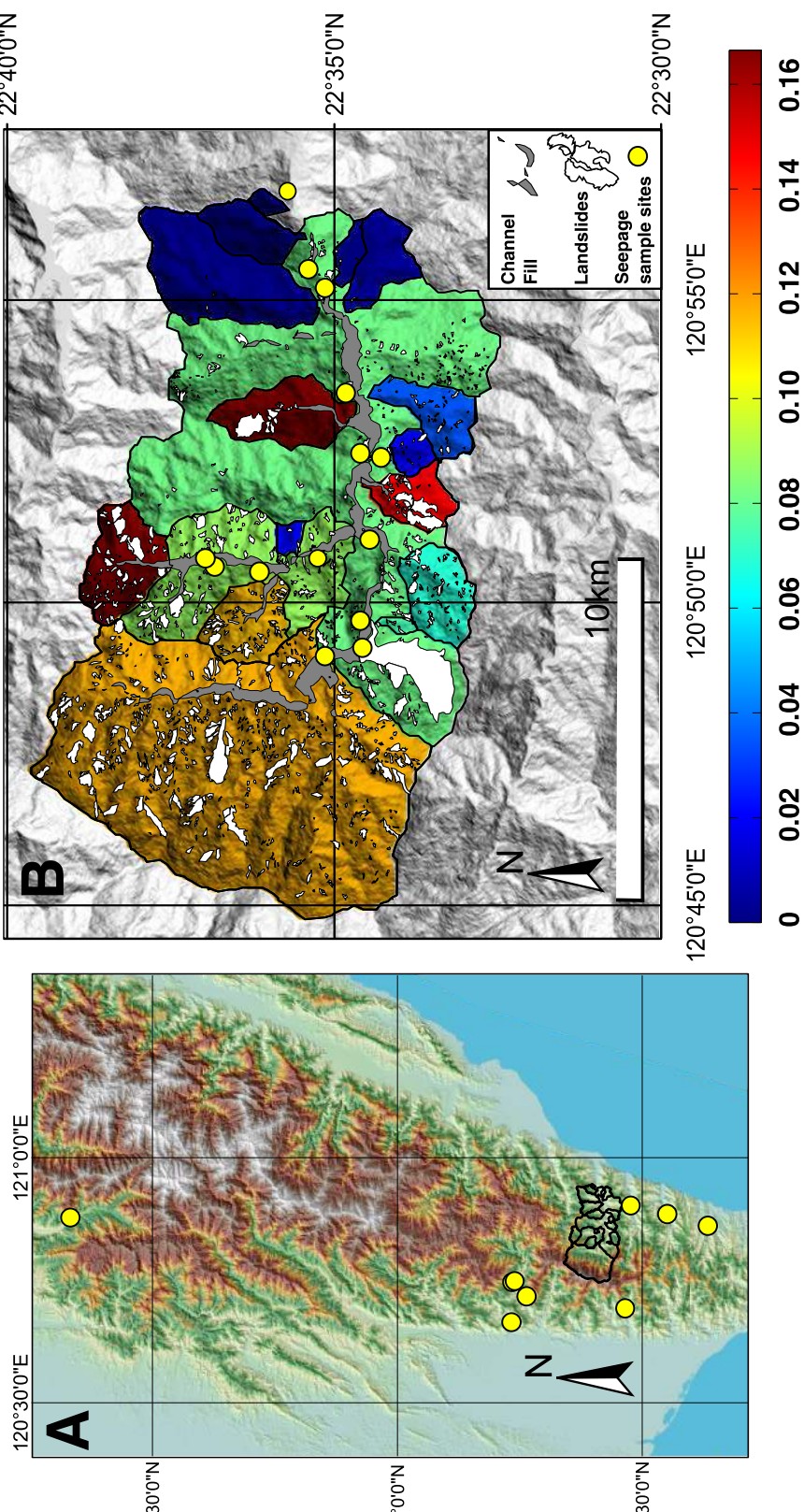

Channel Fill

Landslides

Seepage sample sites

10km

Landslide Fraction

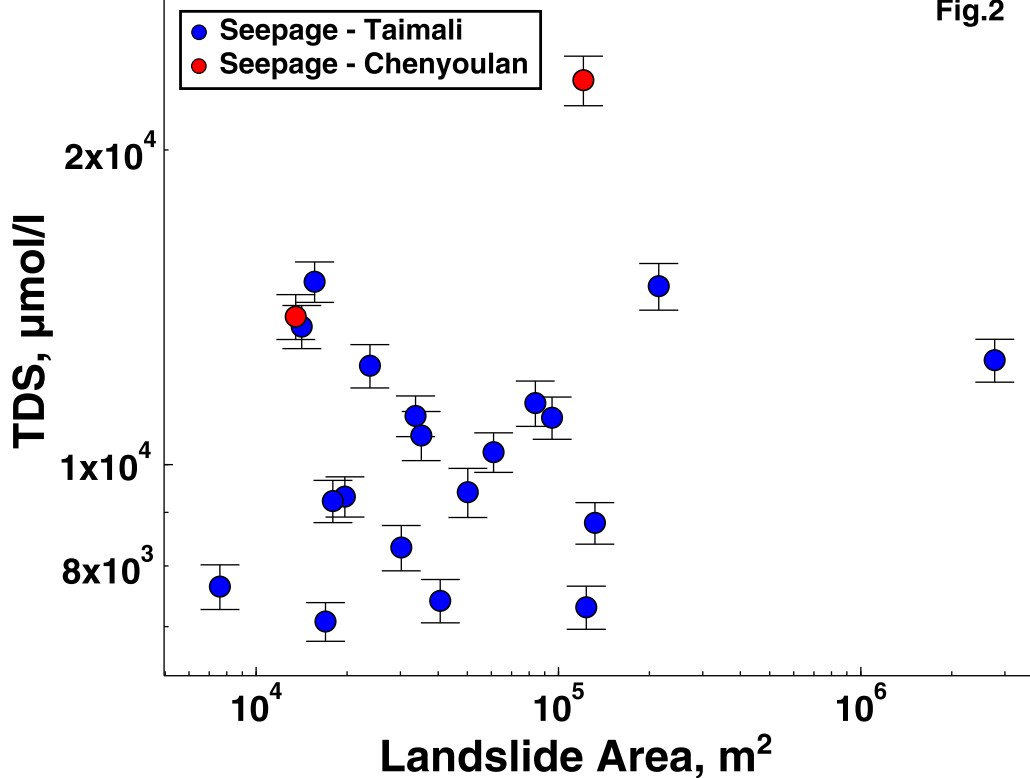

Fig.2

Fig.3A

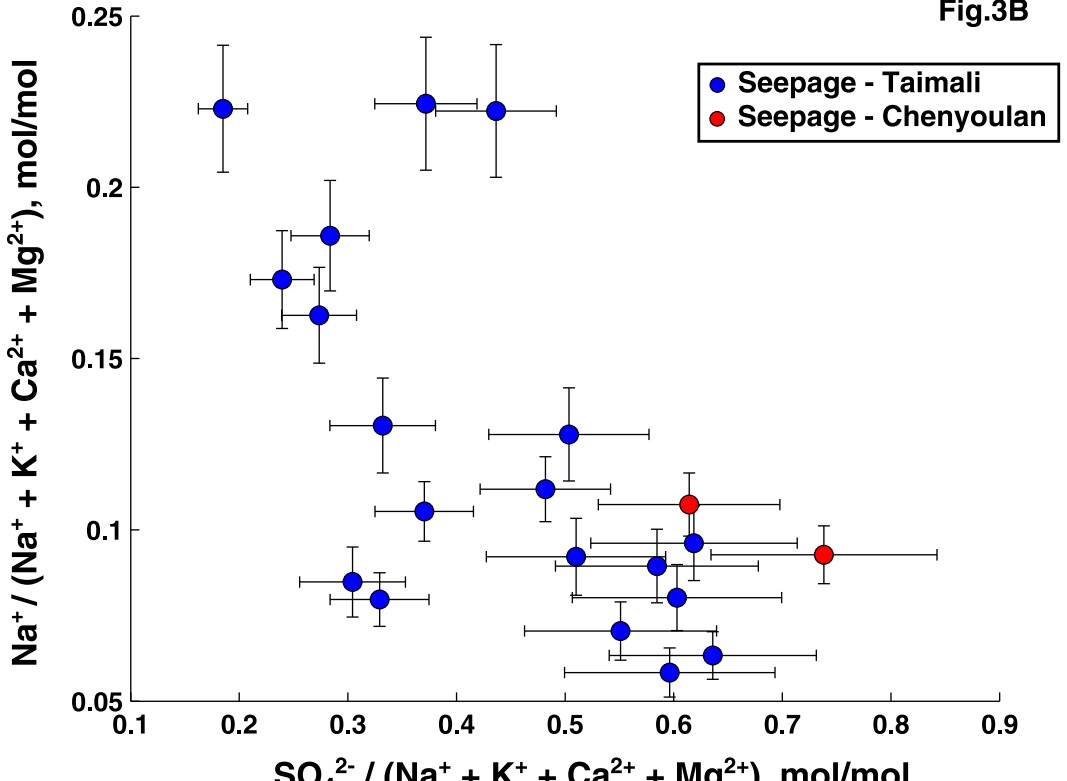

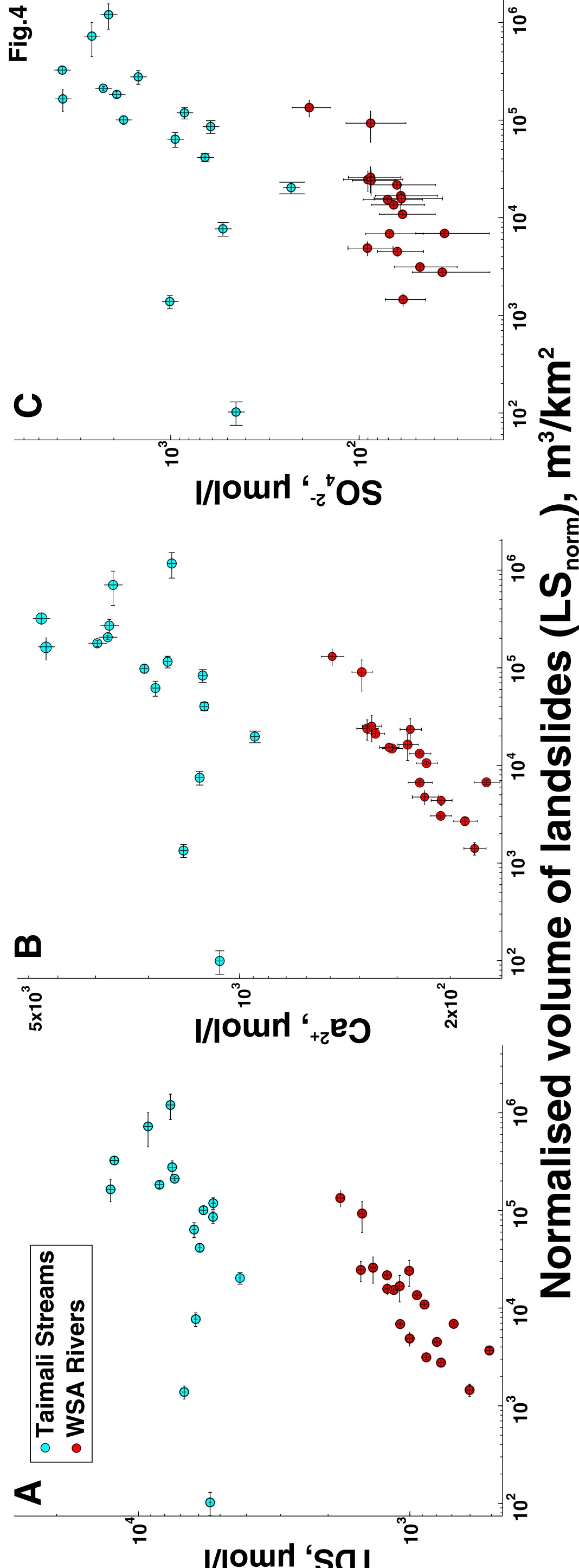

Fig.4

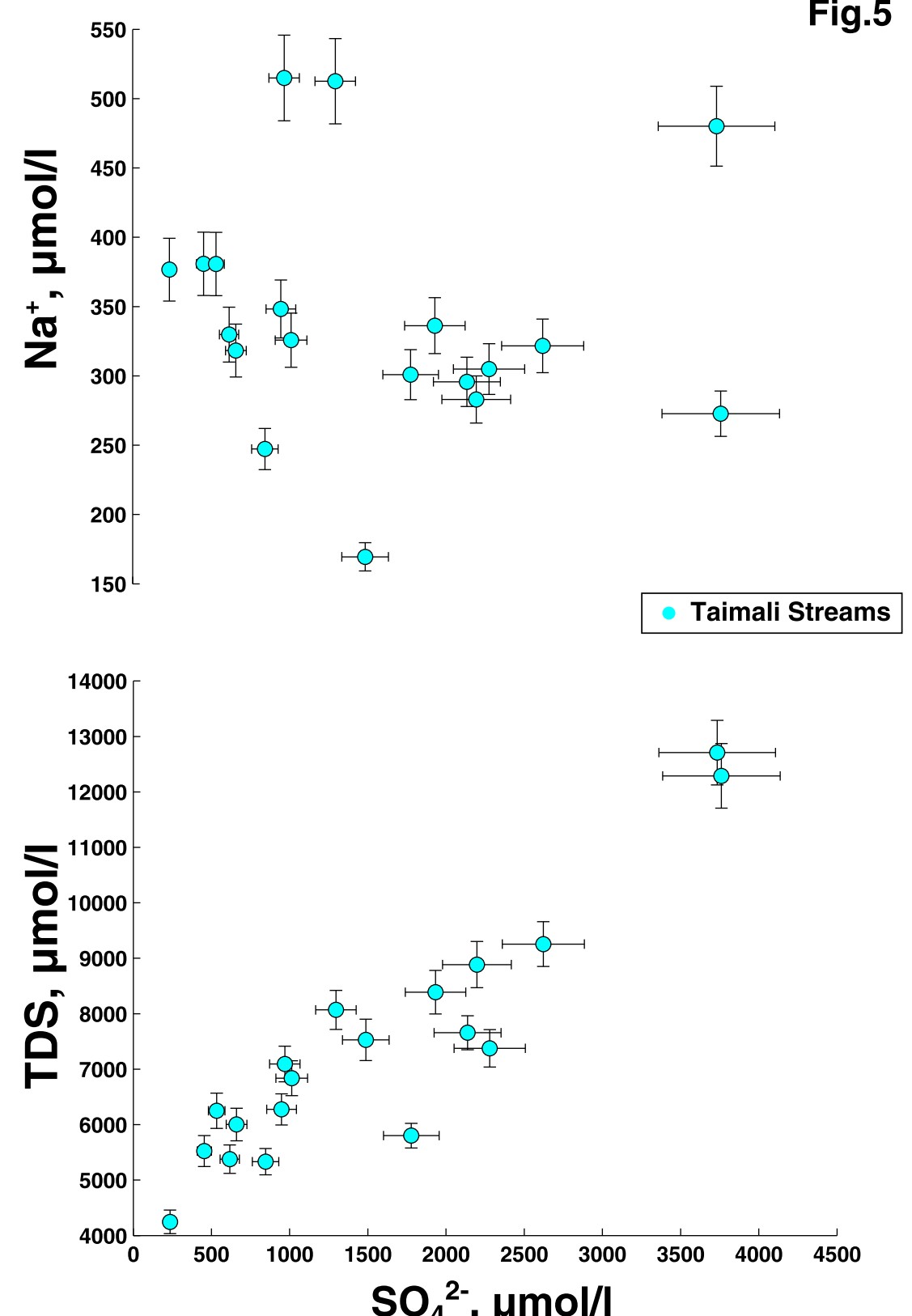

Fig.5