# Peer review of "Oxidation of sulphides and rapid weathering in recent landslides"

_Earth Surface Dynamics, 2016_

## Referee Comment (RC1) · Anonymous Referee #1 · 4 Jul 2016

In their manuscript entitled "Oxidation of sulfides and rapid weathering in recent landslides", Emberson et al report data on the solute geochemistry and size of landslide deposits in the mountains of Taiwan. Exploiting the fact that a majority of the exposed landslides were triggered during a single storm event, the authors test the links between landsliding and chemical weathering independent of time. While the authors do not find a relationship between the size of landslide deposits and the solute concentrations they generate, they do find that rivers in catchments with more landslides tend to have the higher solute concentrations.

Overall, I think that this manuscript is timely and appropriate for publication in ESurf. However, there are a variety of criticisms that I would like to see addressed. In general, I found many instances where either key assumptions were not made clear or statements were not appropriately backed up by published literature. I've tried to highlight

these instances in my more detailed comments below.

**Line 34 This is hardly an exhaustive list of weathering-erosion models. Additionally, the Dixon and Von Blanckenburg paper does not present a new model of the coupling between weathering and erosion. I'd recommend that the authors cite some of the key papers in this field (included below). Additionally, the authors state say that existing weathering-erosion models do not account for the effects of landsliding. While this broadly true, the issue was previously addressed by Gabet (2007; EPSL), who argued that landsliding did not affect the coupling between weathering and erosion predicted by existing models. Since it seems as if the authors are trying to argue the opposite point, it'd be worth discussing the conclusions of Gabet (2007).**

Gabet, E. J. (2007). A theoretical model coupling chemical weathering and physical erosion in landslide-dominated landscapes. Earth and Planetary Science Letters, 264(1-2), 259–265.

Ferrier, K. L., & Kirchner, J. W. (2008). Effects of physical erosion on chemical denudation rates: A numerical modeling study of soil-mantled hillslopes. Earth and Planetary Science Letters, 272(3-4), 591–599. http://doi.org/10.1016/j.epsl.2008.05.024

Lebedeva, M. I., Fletcher, R. C., & Brantley, S. L. (2010). A mathematical model for steady-state regolith production at constant erosion rate. Earth Surface Processes and Landforms, 35(5), 508–524. http://doi.org/10.1002/esp.1954

West, A. J. (2012). Thickness of the chemical weathering zone and implications for erosional and climatic drivers of weathering and for carbon-cycle feedbacks. Geology, 40(9), 811–814. http://doi.org/10.1130/G33041.1

Li, D. D., Jacobson, A. D., & McInerney, D. J. (2014). A reactive-transport model for examining tectonic and climatic controls on chemical weathering and atmospheric $CO_2$ consumption in granitic regolith. Chemical Geology, 365, 30–42. http://doi.org/10.1016/j.chemgeo.2013.11.028

**Line 45 You provide no citations for mineralogical abundances at the different sites. Is this something you measured yourself? If not, you should provide the appropriate citations.**

**Line 80 You make the distinction between landslides receiving water directly from rainfall or from runoff into the crest of the landslide. However, in all figures and tables, it is not made clear which type of landslide each sample corresponds to. I think this distinction would be very helpful since it is reasonable to expect runoff to have higher initial solute concentrations then rainfall.**

**Line 98 What is thick? Is there anyway to put an approximate scale on soil depths in this system?**

**Line 105 These statements seem into imply that all deep groundwater is geothermal in origin, which I do not think is true (e.g., I don't think the groundwaters in Calmels et al 2011 EPSL are geothermal). Additionally, the authors state that samples are corrected for geothermal inputs but never describe how this is done.**

**Line 108 It'd be interesting to know what percentage of all landslides in the catchment were visited during your fieldwork. This would help readers assess whether or not the observation that half of the visited landslides were dry is likely to be representative of the entire system.**

**Line 141 The authors attempt a correction for atmospheric deposition using chloride concentrations and element/chloride ratios. While this is reasonable, it should be mentioned that geothermal waters, which the authors acknowledge are likely solute sources, are extremely enriched in chloride such that a small geothermal input would have a disproportionate effect on chloride concentrations and preclude the use of their proposed chloride mixing equation to correct for atmospheric deposition.**

Furthermore, while the element/chloride ratios of rainfall are often assumed to be close to those of seawater, it is widely observed that ratios such as SO4/Cl are enriched in

rainfall relative to seawater by a factor of ~2 (Andreae et al. 1990 JGR). This is true even for rain that falls directly over the ocean (e.g., Stallard and Edmond 1981 JGR) and likely arises from marine biogenic aerosols. So, while the authors try and defend their choice of using seawater ratios, I am skeptical that it is a good assumption for all of the major elements.

Considering the overall point of the manuscript, I do no think it is that important that the authors correct their data for atmospheric inputs. By selecting a single rainfall end-member, the authors already assume that the chemistry of rainfall does not vary spatially across their catchment. So instead of assuming that 1) all chloride comes from rain and 2) that rainwaters = diluted seawater, which are both probably wrong, they can just compare their data with the assumption that the contribution from atmospheric deposition is not the key source of variability. Alternatively, the authors could utilize a more rigorous >2 end-member mixing approach (too simultaneously account for geothermal inputs), but that seems beyond the scope of this manuscript.

**Line 173 The authors are missing a citation here. Li et al. (2014; G3) also addressed the issues associated with the automatic mapping of landslides.**

**Line 191 While it is true that the oxidation state of Si is +4 in natural waters, I don't think this qualifies it as a cation. For example, most dissolved sulfur also has a positive oxidation state (+6), but the authors correctly label it as an anion because it is speciated as sulfate in most natural waters. Typically, Si is speciated as silicic acid in natural waters and, because the pKa of silicic acid is high, it is fully protonated at the pH values observed in most rivers. So, maybe it is best to call Si a neutral species.**

**Line 200 The authors state that solute concentrations in their Taiwanese streams are high relative to global rivers, but provide neither an average value nor a citation for the typical concentrations of solutes in global rivers. Maybe the using values from the compilation of Gaillardet et al. (1999 Chem. Geo.) would be useful here.**

**Line 217 (and elsewhere) What is "weathering efficiency"? My interpretation would**

be that weathering efficiency is the same as what other authors call "weathering intensity", which is the mass of rock solubilized by weathering divided by the mass of rock exposed by uplift. If this is what the authors mean by weathering efficiency, then I do not understand how differences in solute concentrations between streams and landslide seepage provide a useful constraint on it. Steams presumably average over different timescales and have different mass fluxes compared to landslide seepage, which could both obscure inferences about the proportion of rock solubilized.

If the authors mean something else, they should perhaps define this term in the text.

**Line 224 The authors argue that the effects of landslide area not evident in their data by reporting R2 values. First, what model are the authors using to describe this relationship? Using an incorrect model could lead to a poor R2 despite there being an underlying relationship. Overall, this would by much easier to assess if the authors showed us a scatter plot of landslide area and TDS. After making one myself, its pretty clear that there is not an obvious correlation.**

Additionally, I am a bit confused how the authors get negative R2 values. Unless the authors are calculating some sort of adjusted R2 and just not telling us, they shouldn't be able to get negative values.

**Line 226**

The authors report a correlation between sulfate and TDS and argue that this is evidence for the effects of sulfide mineral oxidation. Such a correlation could arise for any number of reasons and is observed globally (e.g., the data of Gaillardet et al.1999 Chem. Geo. show this relationship).

In general, solute concentrations are all negatively correlated with discharge, so variations in discharge alone could produce a correlation between sulfate and TDS. Similarly, solute concentrations should scale with water/rock interaction time, which could also give you a correlation between TDS and sulfate. Dilution by rainfall, evaporative

concentration, and end-member mixing could all also give you a relationship between sulfate and TDS. And finally, as the authors acknowledge, the dissolution of evaporites could also be responsible for this correlation.

To me, the fact that the catchment is underlain by marine shales is a strong argument for sulfate being sourced from sulfide mineral oxidation. Personally, I do not see the additional insight gained from reporting a correlation between TDS and sulfate concentrations, which is expected (and observed) in most river systems independent of sulfide mineral oxidation.

Personally, I would find a scatter plot comparing elemental ratios much more convincing if the authors are trying to make the point that sulfide mineral oxidation coupled to carbonate dissolution is the dominant chemical reaction occurring within the landslide deposits. For example, SO4 / (Na+K+Ca+Mg) versus Ca / (Na+K+Ca+Mg) would show what proportion of the cationic and anionic budgets are sourced from the weathering products of coupled sulfide mineral oxidation and carbonate dissolution.

**Line 235 The authors attribute the fact that solute concentrations are independent of landslide area to a heterogenous distribution of pyrite. Since landslide areas and volumes vary by 3 to 4 orders of magnitude respectively, does this imply that bedrock pyrite concentrations also vary by 3 to 4 orders of magnitude?**

Alternatively, could it be that bedrock pyrite concentrations are more or less the same, but water fluxes co-varies with landslide area so that water/rock ratios are approximately constant? Or, what if sulfide mineral oxidation is limited by the diffusion of oxygen into the landslides? In this scenario, the lack of correlation between landslide area and sulfate concentration arises from the fact that pyrite in deep landslide deposits is essentially inaccessible to weathering. I suppose that this is what the authors mean by "the degree of its exposure for oxidation", but it might be useful to develop this idea further.

**Line 241 The say that landslides "remove the supply limit" on weathering. Conceptually, this does not make sense to me. To me, supply-limited weathering refers to conditions where all weatherable-material exposed by uplift is dissolved before being exported by erosion. This contrasts with kinetically-limited weathering, where un-weathered material is exported by erosion due to material residence times being shorter than the time required for complete reaction. Ferrier et al. (2016 G3) have a good discussion about these definitions.**

Based on these definitions, landslide weathering can be either supply-limited or kinetically-limited depending on how quickly landslide deposits are transported to the river channel and exported from the catchment. Since the authors do not constrain the residence time of landslides deposits with their study catchment, I am not sure how they are able to discuss supply- versus kinetically-limited weathering in landslide deposits.

**Line 246 Silicon, and to a lesser extent Ca, are not conservative in natural waters (e.g., Garrels and Mackenzie 1967, Jacobson et al. 2002 GCA, and many more), so I do not necessarily see how Si/Ca can be used as a reliable proxy for the proportion of cations sourced from silicate weathering. Furthermore, all of the reported data have elevated Na to Si ratios relative to common silicate minerals (e.g., albite), which is consistent with Si loss due to secondary silicate mineral precipitation (i.e. non-conservative Si behavior).**

**Line 255 While I think I agree with the authors, I find the wording in this section unclear. If I understand correctly, the authors are trying to argue that sulfate is sourced predominantly from in-situ sulfide mineral oxidation as opposed to geothermal inputs. If one assumes that their single sample of geothermal water is representative of all geothermal inputs, then seepage Li concentrations can be used to place an upper bound on potential sulfate contributions from geothermal waters. Assuming all Li is sourced from geothermal waters, the observed Li enrichments in some of the seepage samples would be insufficient to account for a bulk of the sulfate budget due to the low $SO_4$/Li ratio measured is in the single geothermal sample. So, to me, Li concentrations**

not "preclude" geothermal inputs, but instead constrain their contributions to the sulfate budget.

**Line 258 I suppose this comment is similar to the one above. If one assumes that potential groundwater inputs would be characterized by high concentrations of both sodium and sulfate, then the limited co-variation between sodium and sulfate concentrations would suggest that variations in sulfate concentrations are not due to mixing with groundwater.**

While this seems like a reasonable argument, it'd be better supported with some scatterplots showing actual mixing relationships based on elemental ratios (not concentrations). The authors could even use literature data to better constrain some of the end-members.

**Line 270 Does it matter that you are comparing solute concentrations from catchments with different surface areas?**

**Line 276 If sulfide oxidation is limited by O2 diffusion, the weathering of silicates could be de-coupled from the oxidation of sulfides. In this case, a correlation between Na and sulfate would not be expected. And, as stated above, variations in Si concentrations are strongly influenced by secondary mineral precipitation which affects neither Na nor sulfate.**

**Line 305 The authors state that the maximum solute concentrations observed in the landslides are 15 to 20 mM. While I don't disagree with this, I am not sure how this observation alone provides insight into the "saturation" concentration of landslide weathering. While there probably is a maximum possible concentration, it need not be the highest concentration observed in this dataset. Also, saturation might be a poor word choice in this context since it has thermodynamic implications.**

**Line 319 The authors state that the amount of landsliding in each catchment is not correlated with the catchment slope. Where does this observation from from?**

[Figure]

**Line 331 Lee et al. (2015 Env. Sci. Poll. R.) also report concentration-discharge relationships for Taiwanese rivers.**

**Line 336 Tipper et al (2006 GCA) report solute concentration-discharge relationship for Himalayan rivers and Torres et al. (2015 GCA) specifically link mountain catchments with "chemostasis" in the Andes/Amazon.**

**Line 345 Again, I am not sure how a correlation between TDS and sulfate can be used to argue that solutes in stream water are sourced predominantly from landslides. This correlation is not unique to landslides and could arise from many different processes.**

**Line 335 The statement "supersaturated everywhere with respect to Ca" doesn't make sense. Waters cannot be supersaturated with respect to an element. They can, however, be supersaturated with respect to a mineral phase.**

**Line 359 The authors define the term "non-equilibrium weathering" to describe landslide weathering because "the depletion rates of the various minerals involved are likely to be very different."**

I have a few issues with this statement. First, the fact that weathering occurs in the first place is because primary minerals are out of equilibrium with respect to surface conditions. So, by definition, all weathering is non-equilibrium. Second, chemical weathering in any setting will involve minerals that react at different rates.

That said, I agree that landslide weathering is somehow "different". To me, the important difference is that landslides rapidly transport weatherable material from depth to the surface. In contrast, "hillslope" weathering involves the much slower transport of material from depth. Rapid transport of material from depth allows minerals that typically have deep weathering fronts (e.g., pyrite) to be exposed to weathering at the surface.

**Line 393 There definitely needs to be some citations for the links between weathering and atmospheric pCO2. Many of the papers cited elsewhere in the text should also be cited here. However, the authors could also consider citing:**

Lerman, A., Wu, L., & Mackenzie, F. T. (2007). CO2 and H2SO4 consumption in weathering and material transport to the ocean, and their role in the global carbon balance. Marine Chemistry, 106(1-2), 326–350. http://doi.org/10.1016/j.marchem.2006.04.004

Torres, M. A., West, A. J., Clark, K. E., Paris, G., Bouchez, J., Ponton C., Fakirs S.J., Galy, V., Adkins, J. F. (2016). The acid and alkalinity budgets of weathering in the Andes-Amazon system: Insights into the erosional control of global biogeochemical cycles, Earth Planet. Sci. Lett. http:/doi.org/10.1016/j.epsl.2016.06.012

**Line 400 The authors mention a "feedback between drawdown of CO2 and erosion." Since erosion rates are not necessarily modulated by atmospheric pCO2, I do not think this can be accurately called a feedback. Additionally, the authors should probably cite some of Maureen Raymo's work on this topic.**

**Line 404 Should probably include the relevant citations for the links between sulfide mineral oxidation and erosion rates.**

**Line 417 The authors state that the area affected by Typhoon Morakot is now a CO2 source but provide no evidence for this. The affect of weathering on atmospheric pCO2 depends on the precise balance between acid producing and acid consuming reactions (e.g., Torres et al. 2016 EPSL), which was not appropriately addressed in this manuscript.**

**Line 448 It might be worth citing some of the other work on carbonate versus silicate weathering in New Zealand. For example,**

Jacobson, A. D., Blum, J. D., Chamberlain, C. P., Craw, D., & Koons, P. O. (2003). Climatic and tectonic controls on chemical weathering in the New Zealand Southern Alps. Geochimica et Cosmochimica Acta, 67(1), 29–46.

Moore, J., Jacobson, A. D., Holmden, C., & Craw, D. (2013). Tracking the relationship between mountain uplift, Silicate weathering, And long-term CO2 consumption with Ca isotopes: Southern Alps, New Zealand. Chemical Geology, 341, 110–127. http://doi.org/10.1016/j.chemgeo.2013.01.005

**Line 455 Again, stochastic landsliding effects were previously incorporated into a weathering-erosion model by Gabet (2007 EPSL). This work should be mentioned and discussed here.**

**Line 464 Again, the link between erosion and weathering the authors describe is not necessarily a feedback. The rates of tectonic uplift do not necessarily depend upon pCO2, so changes in weathering would not necessarily feedback on erosion rates.**

While I am aware that there are some possible weathering-erosion feedbacks (e.g., through rock strength and/or glaciation), I do no think that the authors are referring to these in this manuscript.

**Table 1. Was DIC measured directly? If so, how? If this column is actually charge balance, then it should be labeled as such. Charge balance does not necessarily equal DIC because of species like carbonic acid and organic bases.**

Also, it seems curious that sample TWS15-91 has no anions other than Cl and SO4 yet has a pH of 8.19. I tried to re-produce this pH with the reported solute concentrations using PHREEQC and couldn't get it to work. That said, I did not try very hard. Anyways, it might be worth looking into that sample a bit more to make sure there are not any analytical issues.

**References Andreae, M., Talbot, R., Berresheim, H., & Beecher, K. (1990). Precipitation chemistry in central Amazonia. Journal of Geophysical Research, 95(89), 16987–16999.**

Calmels, D., Galy, A., Hovius, N., Bickle, M., West, A. J., Chen, M., & Chapman, H. (2011). Contribution of deep groundwater to the weathering budget in a rapidly eroding mountain belt, Taiwan. Earth and Planetary Science Letters, 303(1-2), 48–58.

http://doi.org/10.1016/j.epsl.2010.12.032

Ferrier, K. L., & Kirchner, J. W. (2008). Effects of physical erosion on chemical denuda-
tion rates: A numerical modeling study of soil-mantled hillslopes. Earth and Planetary
Science Letters, 272(3-4), 591–599. http://doi.org/10.1016/j.epsl.2008.05.024

Ferrier, Ken L., Clifford S. Riebe, and W. Jesse Hahm. (2016). Testing for sup-
ply‐limited and kinetic‐limited chemical erosion in field measurements of regolith
production and chemical depletion. Geochemistry, Geophysics, Geosystems.

Gabet, E. J. (2007). A theoretical model coupling chemical weathering and physi-
cal erosion in landslide-dominated landscapes. Earth and Planetary Science Letters,
264(1-2), 259–265.

Gaillardet, J., Dupré, B., Louvat, P., & Allegre, C. J. (1999). Global silicate weather-
ing and CO2 consumption rates deduced from the chemistry of large rivers. Chemical
Geology, 159(1-4), 3–30. Garrels, R., & Mackenzie, F. (1967). Origin of the chemi-
cal compositions of some springs and lakes. Equilibrium Concepts in Natural Water
Systems, 222–242.

Jacobson, A. D., Blum, J. D., & Walter, L. M. (2002). Reconciling the elemental and
Sr isotope composition of Himalayan weathering fluxes: insights from the carbonate
geochemistry of stream waters. Geochimica et Cosmochimica Acta, 66(19), 3417–
3429.

Jacobson, A. D., Blum, J. D., Chamberlain, C. P., Craw, D., & Koons, P. O. (2003).
Climatic and tectonic controls on chemical weathering in the New Zealand Southern
Alps. Geochimica et Cosmochimica Acta, 67(1), 29–46.

Lebedeva, M. I., Fletcher, R. C., & Brantley, S. L. (2010). A mathematical model for
steady-state regolith production at constant erosion rate. Earth Surface Processes
and Landforms, 35(5), 508–524. http://doi.org/10.1002/esp.1954

Lee, T.-Y., Hong, N.-M., Shih, Y.-T., Huang, J.-C., & Kao, S.-J. (2015). The

sources of streamwater to small mountainous rivers in Taiwan during typhoon and non-typhoon seasons. Environmental Science and Pollution Research. http://doi.org/10.1007/s11356-015-5183-2

Lerman, A., Wu, L., & Mackenzie, F. T. (2007). CO2 and H2SO4 consumption in weathering and material transport to the ocean, and their role in the global carbon balance. Marine Chemistry, 106(1-2), 326–350. http://doi.org/10.1016/j.marchem.2006.04.004

Li, D. D., Jacobson, A. D., & McInerney, D. J. (2014). A reactive-transport model for examining tectonic and climatic controls on chemical weathering and atmospheric CO2 consumption in granitic regolith. Chemical Geology, 365, 30–42. http://doi.org/10.1016/j.chemgeo.2013.11.028

Li, Gen, et al. (2014). Seismic mountain building: Landslides associated with the 2008 Wenchuan earthquake in the context of a generalized model for earthquake volume balance. Geochemistry, Geophysics, Geosystems 15.4, 833-844.

Moore, J., Jacobson, A. D., Holmden, C., & Craw, D. (2013). Tracking the relationship between mountain uplift, Silicate weathering, And long-term CO2 consumption with Ca isotopes: Southern Alps, New Zealand. Chemical Geology, 341, 110–127. http://doi.org/10.1016/j.chemgeo.2013.01.005

Stallard, R., & Edmond, J. (1981). Geochemistry of the Amazon 1. Precipitation chemistry and the marine contribution to the dissolved load at the time of peak discharge. Journal of Geophysical Research, 86(C10), 9844–9858.

Tipper, E. T., Bickle, M. J., Galy, A., West, A. J., Pomiès, C., & Chapman, H. J. (2006). The short term climatic sensitivity of carbonate and silicate weathering fluxes: Insight from seasonal variations in river chemistry. Geochimica et Cosmochimica Acta, 70(11), 2737–2754.

Torres, M., Joshua West, A., & Clark, K. E. (2015). Geomorphic regime modulates hydrologic control of chemical weathering in the Andes-Amazon. Geochimica et Cosmochimica Acta, 166, 105–128. http://doi.org/10.1016/j.gca.2015.06.007

Torres, M. A., West, A. J., Clark, K. E., Paris, G., Bouchez, J., Ponton C., Fakirs S.J., Galy, V., Adkins, J. F. (2016). The acid and alkalinity budgets of weathering in the Andes-Amazon system: Insights into the erosional control of global biogeochemical cycles, Earth Planet. Sci. Lett. http:/doi.org/10.1016/j.epsl.2016.06.012

West, A. J. (2012). Thickness of the chemical weathering zone and implications for erosional and climatic drivers of weathering and for carbon-cycle feedbacks. Geology, 40(9), 811–814. http://doi.org/10.1130/G33041.1

---

## Referee Comment (RC2) · Anonymous Referee #2 · 5 Jul 2016

Emberson et al., Oxidation of sulphides and rapid weathering in recent landslides. Earth Surface Dynamics.

Determining how erosion and weathering are related to one another in tectonicallyactive mountains is important for understanding how mountain uplift influences geochemical cycling. Here Emberson et al. present geochemical data from surface waters in Taiwan. They focus on the chemistry of stream waters as well as waters that have flowed through landslide debris, deposited by extensive hillslope failures associated with Typhoon Morakot. The pyrite and carbonate bearing marine sedimentary rocks in the study area offer a mineralogical contrast to a complimentary study the group conducted in the Southern Alps of New Zealand, where the mineralogy of the bedrock differs. Emberson et al. conclude that sulfuric acid generated by pyrite oxidation drives the dissolution of carbonate rock within the fractured landslide mass and that, above a

threshold minimum of landsliding, that total dissolved solids, Ca, and sulfate increase with increasing catchment landslide volume. These results provide new, process-level insight on the generation of solutes in tectonically-active landscapes, and will be of broad interest and is an appropriate contribution to Earth Surface Dynamics. I believe the manuscript can be accepted for publication following revision to address questions regarding several aspects of the paper.

Main comments: This manuscript lacks a distinct Results section, nor is the statistical treatment of the data clearly described in the Methods section, such that the reader does not know how the various dataset are analyzed with respect to one another until the statistics are presented as part of the Discussion. A revised manuscript without an explicit Results section could work, but if so, additional description of the Methods are needed to better link the different sections of the paper and prepare the reader for the Discussion.

Line 318-319. It is worthwhile to present data that demonstrate that the magnitude of landsliding (and TDS) does not scale with topographic metrics such as slope or relief. The reason being is that erosion rates (and hence coupled weathering rates) scale with catchment topography. Hence it might be expected that, even in the absence of any landsliding, the range of TDS values could be explained by topographic variation of the watersheds. For the arguments presented in the paper, it is important to show this is not the case.

Line 406-414. It would be worthwhile to elaborate and make the logic very clear here, as this paragraph is laying out a conceptual framework, based on the observations, for understanding the role landslides play in the weathering budget of mountains. In particular, more explanation of the role of sediment retention time would be useful. The instantaneous weathering flux from the landslide debris should decay as a function of time, whereas the cumulative flux of weathering products from landslide debris will increase with time. Hence there should be more explanation to support the claim that long sediment residence times would have a limited effect on CO2 budgets-if this

**ESurfD**
indeed is the case, which seems unlikely. The return interval of landslide-triggering events should also play a role, as this ultimately sets the volume per time of landslide debris subjected to sulfuric acid weathering. A conceptual diagram showing the roles of residence time and return interval would be a useful approach to organize these thoughts in this paragraph. Additionally, there has been at least some work on 14C dating of alluvial deposits in southern Taiwan, presumably which record valley alluviation due to landsliding in some cases. Although these studies may not be in the same catchment as was investigated here, these studies could be drawn upon to estimate to a first order the length of time landslide derived sediment is likely to persist in the watershed.

Hsieh, Meng-Long, and Shyh-Jeng Chyi. "Late Quaternary mass-wasting records and formation of fan terraces in the Chen-yeo-lan and Lao-nung catchments, central-southern Taiwan." Quaternary Science Reviews 29.11 (2010): 1399-1418.

Hsieh, Meng-Long, and Peter LK Knuepfer. "Synchroneity and morphology of Holocene river terraces in the southern Western Foothills, Taiwan: A guide to interpreting and correlating erosional river terraces across growing anticlines." GSA Special Paper (2002): 55-74.

Lines 415-420. Here it would be useful to point the way forward in terms of what data are needed to determine the sign of the CO2 budget in a rapidly eroding mountain range. There are already elements of this in the paragraph, but a thorough explanation of what suite of measurements are needed would be a forward looking way to wrap up the manuscript. Additionally, with the understandably limited data that are available, is there at least a signal of what process is dominating the CO2 budget, sulfuric acid weathering of carbonate, carbonic acid weathering of silicates, or organic carbon burial? For example, back of the envelope calculations assuming regional-scale runoff values to infer fluxes for this study, and comparison with previous work on organic carbon export in Taiwan may be useful, even if imprecise. Even if a very first order budget cannot be calculated, due to lack of constraints in the values for the various parts of the

**ESurfD**
budget, indicating that this is the case is a useful thing to do, as it points to the need for a different strategy in approaching erosion-CO2 links in mountains.

**tains. ESurfD**
On a similar note, the paper begins by asking a question regarding the general influence of landslides on weathering and roles lithology and climate may play (Line 42-44). In concluding the Discussion it would be worthwhile to return to the specific parts of that question and assess the state of knowledge, as informed by this work, and prior publications.

Line 427-429 (also line 241). There is a contradiction in this statement, in that the suggestion is that weathering is kinetically-limited, so long as the involved minerals are 'abundantly exposed'. The 'abundant exposure' is suggestive of supply limited weathering, and one interpretation of figure 3 (excluding the smaller landslide volumes in Taiwan), is that the supply of material by landsliding is setting the concentrations of dissolved species and likely also the weathering flux. My suggestion is to be very explicit here regarding use of the terms kinetic and supply limited weathering. Pyrite oxidation depends on kinetics (as do all reactions), but when the numbers of minerals involved in such reactions are increased, likely by orders of magnitude due to fracturing of the rock mass during landslide runout, then the supply of minerals plays a very large role in setting the concentrations of various dissolved species measured in the water.

Other comments: Line 14. Insert 'landslide' before seepage

Line 18 and elsewhere. Two significant digits on the R2 values is probably sufficient.

Line 34. Several additional citations are warranted here, including Gabet and Mudd (2009), Ferrier and Kirchner (2008) and West, (2012). Also, Dixon and von Blanckenburg do discuss the role of landsliding.

Line 38 and elsewhere. Many citations do not follow a uniform format, with author initials appearing in some cases. Some editing the entries in the reference management software/database that was used here is needed. Line 46. 'local' is ambiguous, perhaps replace with 'landslide'

Line 56. 'importantly' can be omitted in this case and elsewhere in the manuscript. Alternatively, switch the order of 'importantly' and 'propagate'

Line 79-81. Perhaps break this into two sentences or put parentheses around: either directly from rainfall.....crest of landslides.

Line 90-91. This statement ought to have a citation to support it.

Line 98. While I have not been to the field site, observations elsewhere in Taiwan would suggest, based on the continuity of vegetation coverage, that a soil (or mobile regolith) mantle, while not necessarily laterally continuous, must cover a large fraction of the bedrock on hillslopes.

Line 105. Perhaps change 'in view of this' to 'in light of this'

Line 115. The term 'channel fill' has an ambiguous meaning in this sentence. Does it have the same meaning as 'alluvium'? Some clarification is needed.

Line 165. 'landslides' should be singular

Line 169. Perhaps edit 'as prescribed by' to 'as described in'

Line 173. A G-cubed paper by Gen Li and colleagues also provides an important example of the effects of amalgamating landslide areas. Li, Gen, et al. "Seismic mountain building: Landslides associated with the 2008 Wenchuan earthquake in the context of a generalized model for earthquake volume balance." Geochemistry, Geophysics, Geosystems 15.4 (2014): 833-844.

Line 197. Edit so that 'sit at' is removed

Line 201. The statement that TDS values are high with respect to global concentrations needs a citation.

Line 220-221. Whether deeper landslides will expose proportionally more calcite to
weathering relative to shallow landslides will depend on the depth of the calcite weathering front. Whereas much calcite is likely lost from the soil, the soils are likely thin (

Line 329. First two words have unclear meaning.

Line 334. Replace 'in keeping' with consistent

Line 336. Problems with these references. e.g., should be Lyons et al.

Line 348. A 'likely' should be inserted between 'exhumation' and 'not'. There are no data on the distribution of sulfide minerals within the rock, so you can only hypothesize that this variability also controls the variability in seepage concentrations at individual landslides.

Line 357-358. Suggest editing to: '...and the associated generation of sulfuric acid.'

Line 360. Different than what? Please complete this thought.

Line 361. Is there a better term than 'minority' minerals - reactive, highly soluble, labile? Choose one and use it consistently throughout. For example, there are likely many phases present in low abundance that do not contribute to non-equilibrium weathering.

Line 368. Some sort of punctuation to separate the sample numbers and the values of the measurements would be helpful.

Line 376. The statement regarding weathering by carbonic acid dominating in other parts of the landscape should be qualified. It is likely the case, but data are not discussed that support this. Can the data from the rivers be used to support this inference?

Line 378. It is not clear what 'restricted' means in this context. Please clarify.

Line 381. There is a comma that can be omitted.

Line 403. Elaborate on '...a more central role,...' It will be useful to be explicit regarding what the central role refers to.

Line 406. Suggest replacing the 'As' with 'when'

Line 426. Again, unclear if the waters in the landslide debris indeed circulate.
Line 430. Omit 'in the first instance' or replace with another term, such as foremost.

Line 434. Replace 'units' with 'rocks' and replace 'This' with a more explicit description of the process.

Line 436. Meaning of 'subsumed' is unclear in this context. Some editing is needed.

Line 339. Omit 'first' or replace with another term than does not imply a temporal relationship.

Line 454. 'mineralogical' instead of 'lithological'?

Line 457. Gabet 2007, EPSL is a study that has examined the stochastic role of landslides in weathering.

Figure captions. 'Total dissolved load (TDS)' – the acronym should match the first letters of each word, e.g., total dissolved solids or TDL.

Figure 1. It would be useful to have an inset map showing Taiwan with a box showing the area of Figure 1a. This inset could go in the lower right corner of Fig. 1a.

Figure 2. There should be at least two labels on the y-axis in each panel to easily allow the reader to see the data range.

**ESurfD**

---

## Referee Comment (RC3) · Anonymous Referee #3 · 12 Jul 2016

The authors argue that landslides place unweathered material at the land surface, exposing pyrite to oxidation which drives weathering. They discuss many implications.

The paper was interesting and this phenomenon deserves attention because it is important. I have seen dissolved Fe emanating from beneath landslides in Puerto Rico (and soon thereafter precipitating), right after a landslide, so I know this happens and the central idea is not surprising. Pyrite oxidizes fast when exposed to oxygen and water and especially when bacteria are involved. So that is not that surprising. I think what the authors are trying to do that is interesting is quantify this effect by looking at different watersheds with different densities of landslides. This is an interesting idea. But I don't think they quite nailed down their story. I have a hard time even knowing exactly what they did show, and what this means exactly. I suggest that the authors work hard to probe their data more. With that effort, their results will be more impactful.

Part of the problem is that much of the phrasing is a bit vague and confusing. I got tired trying to figure out what they meant. Even the abstract has such phrasing. For example, this phrasing in the abstract is awkward and unclear and should be revised: "Bedrock landslides create conditions for weathering where all mineral phases in a lithology are initially unweathered within landslide deposits, and therefore the most labile phases dominate the weathering at the outset and during a transient period of depletion." I can't really read the abstract and get the main point. Given that I am not surprised about pyrite oxidation being accelerated by landslides...what is the big take-home message that they can show? One interesting aspect that is not explained clearly is why Taimali operates differently than WSA: shouldn't this basic difference be in the abstract?

I do think the authors should rework their paper before final publication. Here are some points of various importance to consider:

1. The key question in my mind is the depth to which landslides scour the landscape in comparison to the depth of the pyrite depletion zone and the carbonate depletion zone. The authors sort of mention this but not fully. Where are the reaction fronts for pyrite and carbonate in these rocks? And what is the depth to which landslides scour? Often pyrite is oxidized down to the water table.

2. Please discuss the calculation of bicarbonate by charge balance. This would usually have a large or reasonably large error just because it is a small difference between larger numbers. But, for example, the calculation could also have error due to neglect of organic anions in the waters, or any other anions. What is the error in bicarbonate? (and it should not be labelled DIC, it should be labelled bicarbonate). I see that the measurements of sulfate and chloride are each +_10%...seems like the calculation of bicarbonate by difference might have big error bars. In that regard, there are most likely too many significant figures on the bicarbonate (and probably some of the other element concentrations?)

3. The calculation of bicarbonate was then used to assess calcite supersaturation. I would think there is a big error in this calculated SI as well.

4. I wouldn't capitalize element names (Calcium versus calcium).

5. Generally, I would not call Si "cationic load" although we do usually refer to it as a cation simply because we analyze it with other cations. But it is present as a neutral species unless the pH is very high. For this reason, Si does not contribute to cation balance.

6. With respect to Sr: what does this mean? "peaks at 30 umol/L

7. The paper has a lot of assertions that are really undefended assumptions. Such assertions need to be defended more thoroughly: "the depth of a landslide scales with the area"…is this always true for all locations? or was Larsen reporting about one locality more than another? I am also not sure about "the proportion of calcite available correlates with the size of the issuing landslide". I think this statement could be true but doesn't have to be true. Are there data corroborating this assumption? The authors later state that "the lack of correlation between landslide size and seepage concentration suggests that the distribution of pyrite is highly heterogeneous.." I am not sure I would say the pyrite distribution is heterogeneous but I would argue that in many places there could be a pyrite oxidation front like those identified in Chigira et al. 1990; 1991; Drake et al., 2009; Brantley et al., 2013. I suppose that the base of landsliding in any given area might be somehow correlated with the depth to pyrite, but I think it is also very possible that the depth of landsliding might not correlate with the depth to pyrite.

8. The paper needs some mineral abundance data and element abundance data.

9. The authors start using the phrase "Landslide weathering efficiency" without defining it. What exactly is this?

10. Please explain why you are using Kendall's tau value?

11. Line 243, acidity means the base neutralizing capacity of a water. Do the authors mean acidity or protons? (Pyrite is a source of protons.)

12. I suggest the authors include some sort of figure or more text to defend this statement: "the lower proportion of purely silicate derived cations in landslide seepage . . .supports the interpretation that weathering in the sampled landslides is dominated by the dissolution of carbonates".

13. I don't understand this: ". . .a significant proportion of the sedimentary pyrite contained within the rock mass survives exhumation. . ." I think in general, pyrite moves up and out of the system like any other mineral until it gets oxidized at some depth related to the rate of erosion and the amount of water. Is this is what is meant by survival (death by oxidation)? It is generally oxidized down to the water table unless it is a very pyrite rich rock.

14. The authors argue that Na and Si do not vary with sulfate but Ca and K do vary with sulfate. They argue that this points to multiple pathways relevant for silicate weathering on a catchment scale. I don't see how an observation about landslide seepage can tell us about the other pathways: it can only tell us about the landslide contribution, not the other contributions. I think there are nested reaction fronts. . .deeper flowpaths will interact with parent lithology and shallower flowpaths will not. The authors are arguing that landslides bring some of the deeper material to the surface: landslides are egg beaters. What we need to know is, how deep do they scour relative to where the minerals are?

15. I think this is unsupported as written: "the relationship between landslide volume and stream solute concentration will also saturate above a hypothetical maximum concentration"

16. Waters cannot be supersaturated with respect to an element. . .only wrt a mineral (unless we are talking about native sulfur or gold or some such). See line 356. This needs to be fixed.

17. I don't understand this sentence at all! I think it should be deleted or re phrased. "We describe this style of weathering as non equilibrium weathering as the depletion rates of the various minerals involved are likely to be very different."

18. What is important is not the "minority minerals" but "reactive minerals". Once the reactive minerals are removed, then the landslide material will weather like all the other weathered material in the landscape (i.e. material above the reaction front for the reactive mineral)...unless the landslide is grinding up the material and increasing surface area, which is another possible effect. I do think that pyrite is often a minor mineral but that is not what is important about it.

19. Once pyrite oxidizes, weathering will revert from H2SO4 to carbonic plus organic acids.

20. The authors' use of "ambivalent" is incorrect: "the impact of landsliding on climate is ambivalent". I think the authors mean ambiguous.

21. I am not sure that the authors can defend the statement, "the area most affected by erosion due [to] typhoon Morakot is now a net source of CO2"

22. Conclusions: I don't understand this sentence: Five years after landsliding, carbonic acid is a weathering agent of lesser importance and the weathering of silicate mienrals is subsumed.

23. The term "lithological phases" is not correct. A lithology is a lithology and a phase is a phase...not sure what a lithological phase is...a mineral?

24. In figure 4, SiO2 is silica, Si is silicon.

25. The way the figure on page 20 is plotted is confusing. The fonts on the y axis differ above and below the break. I don't really understand what the tick marks mean on the y axis, because of the break in the axis. Why plot the two figures together in this way?

---

## Author Comment (AC1) · 3 Aug 2016

The comment was uploaded in the form of a supplement:
http://www.earth-surf-dynam-discuss.net/esurf-2016-31/esurf-2016-31-AC1-supplement.pdf

---

## Author Response (AR1)

**Authors Response to Reviews of 'Oxidation of sulphides and rapid weathering in recent landslides'**

*We are grateful for three thorough and helpful reviews, which have helped improve the manuscript significantly. In the following responses, the original comments are kept in full, and our responses and actions are italicised.*

**Anonymous Referee #1**

In their manuscript entitled "Oxidation of sulfides and rapid weathering in recent landslides", Emberson et al report data on the solute geochemistry and size of landslide deposits in the mountains of Taiwan. Exploiting the fact that a majority of the exposed landslides were triggered during a single storm event, the authors test the links between landsliding and chemical weathering independent of time. While the authors do not find a relationship between the size of landslide deposits and the solute concentrations they generate, they do find that rivers in catchments with more landslides tend to have the higher solute concentrations.

Overall, I think that this manuscript is timely and appropriate for publication in Esurf. However, there are a variety of criticisms that I would like to see addressed. In general, I found many instances where either key assumptions were not made clear or statements were not appropriately backed up by published literature. I've tried to highlight these instances in my more detailed comments below.

**Line 34 This is hardly an exhaustive list of weathering-erosion models. Additionally, the Dixon and Von Blanckenburg paper does not present a new model of the coupling between weathering and erosion. I'd recommend that the authors cite some of the key papers in this field (included below). Additionally, the authors state say that existing weathering-erosion models do not account for the effects of landsliding. While this broadly true, the issue was previously addressed by Gabet (2007; EPSL), who argued that landsliding did not affect the coupling between weathering and erosion predicted by existing models. Since it seems as if the authors are trying to argue the opposite point, it'd be worth discussing the conclusions of Gabet (2007).**

Gabet, E. J. (2007). A theoretical model coupling chemical weathering and physical erosion in landslide-dominated landscapes. Earth and Planetary Science Letters, 264(1-2), 259–265.

Ferrier, K. L., & Kirchner, J. W. (2008). Effects of physical erosion on chemical denudation rates: A numerical modeling study of soil-mantled hillslopes. Earth and Planetary Science Letters, 272(3-4), 591–599. http://doi.org/10.1016/j.epsl.2008.05.024

Lebedeva, M. I., Fletcher, R. C., & Brantley, S. L. (2010). A mathematical model for steady-state regolith production at constant erosion rate. Earth Surface Processes and Landforms, 35(5), 508–524. http://doi.org/10.1002/esp.1954

West, A. J. (2012). Thickness of the chemical weathering zone and implications for erosional and climatic drivers of weathering and for carbon-cycle feedbacks. Geology, 40(9), 811–814. http://doi.org/10.1130/G33041.1

Li, D. D., Jacobson, A. D., & McInerney, D. J. (2014). A reactive-transport model for examining tectonic and climatic controls on chemical weathering and atmospheric CO2 consumption in granitic regolith. Chemical Geology, 365, 30–42. http://doi.org/10.1016/j.chemgeo.2013.11.028

*The reviewer correctly points out that the list of references in not exhaustive. We have added a discussion of the Gabet paper, as that has been an important precursor for both this work and our previous work in the WSA. We have also cited the other papers in the introduction to increase the completeness of our citation.*

**Line 45 You provide no citations for mineralogical abundances at the different sites. Is this something you measured yourself? If not, you should provide the appropriate citations.**

*The missing references (Das et al. 2012 for Taiwan, Chamberlain et al. 2005, Koons and Craw 1991 for the WSA) have been added.*

**Line 80 You make the distinction between landslides receiving water directly from rainfall or from runoff into the crest of the landslide. However, in all figures and tables, it is not made clear which type of landslide each sample corresponds to. I think this distinction would be very helpful since it is reasonable to expect runoff to have higher initial solute concentrations then rainfall.**

*The reviewer raises an important point that runoff is likely to have more concentrated solute contents than rainfall. However, we would suggest that the input of solutes along a flow path is not additive; it is subject to saturation with respect to the weathering minerals. Both rainfall and runoff may be undersaturated with respect to the minerals weathering, but could both become saturated within landslides (we point to evidence of supersaturation with respect to calcite in all seepage samples). We have no constraints on the chemistry of runoff into the landslides from which we have sampled seepage. Moreover, the resolution of available DEMs does not always permit a robust estimation of the drainage area upslope of the landslide crest. For these reasons, we have chosen not to provide detailed information about this issue. Despite this, we have altered the text to resolve confusion.*

*Lines 91-92: "This water has circulated through the landslides and derives from rainfall, and in some cases is likely supplemented by runoff into crests of landslides."*

**Line 98 What is thick? Is there anyway to put an approximate scale on soil depths in this system?**

*The original use of the word 'thick' is subjective, and we have changed it. Lee and Ho (2009) found soils in the mountains of Taiwan >1m thick only in the bottom of valleys. We note that in these steep landscapes the soil thickness in slopes on short (10s metres) length scales varies widely even in more stable catchments.*

*Lines 111-113: "Soils are generally only >1m in local concavities in the landscape (Lee and Ho 2009), but contiguous, dense vegetation on hillslopes that have not been dissected by landsliding suggests sufficient mobile regolith or soil is present to support flora."*

**Line 105 These statements seem into imply that all deep groundwater is geothermal in origin,**

which I do not think is true (e.g., I don't think the groundwaters in Calmels et al 2011 EPSL are geothermal). Additionally, the authors state that samples are corrected for geothermal inputs but
95  never describe how this is done.

*The reviewer is completely correct. We agree that geothermal and deep groundwaters are not identical. We have better constraint on geothermal waters in this system, having sampled them. In the first version of the text we described a correction, but what we meant was that we checked the*
100 *non geothermal samples for elevated Lithium contents; if they are >10umol/l then we would exclude them, as this implied >1% geothermal input (based on the measured lithium concentration in the hot spring sample). None of the samples fit this description though, so none were excluded. We have altered the text to remove mention of correction, and instead stress that the geothermal sample is used to qualitatively check for input from this source to either sulphate or other elements*
105 *in the seepage samples.*
*See lines 123-125: "We compare sampled seepage with our own measurements of geothermal water as well as with the composition of deep groundwater reported in the literature to qualitatively assess the importance of deeper-sourced water in landslide outflow."*

110 ### Line 108 It'd be interesting to know what percentage of all landslides in the catchment were visited during your fieldwork. This would help readers assess whether or not the observation that half of the visited landslides were dry is likely to be representative of the entire system.

*We mapped 576 landslides in the Taimali catchment, so we visited about 10% of these. We have*
115 *included this detail in the text –*
*Lines 129: "despite the dry conditions more than half of around 50 deep-seated landslides we visited had persistent seepage" & 204: "We mapped 576 landslides extant in the Taimali"*

**Line 141 The authors attempt a correction for atmospheric deposition using chloride**
120 concentrations and element/chloride ratios. While this is reasonable, it should be mentioned that geothermal waters, which the authors acknowledge are likely solute sources, are extremely enriched in chloride such that a small geothermal input would have a disproportionate effect on chloride concentrations and preclude the use of their proposed chloride mixing equation to correct for atmospheric deposition. Furthermore, while the element/chloride ratios of rainfall are often assumed
125 to be close to those of seawater, it is widely observed that ratios such as $SO_4/Cl$ are enriched in rainfall relative to seawater by a factor of 2 (Andreae et al. 1990 JGR). This is true even for rain that falls directly over the ocean (e.g., Stallard and Edmond 1981 JGR) and likely arises from marine biogenic aerosols. So, while the authors try and defend their choice of using seawater ratios, I am skeptical that it is a good assumption for all of the major elements.
130 Considering the overall point of the manuscript, I do no think it is that important that the authors correct their data for atmospheric inputs. By selecting a single rainfall end-member, the authors already assume that the chemistry of rainfall does not vary spatially across their catchment. So instead of assuming that 1) all chloride comes from rain and 2) that rainwaters = diluted seawater, which are both probably wrong, they can just compare their data with the assumption that the
135 contribution from atmospheric deposition is not the key source of variability. Alternatively, the authors could utilize a more rigorous >2 end-member mixing approach (too simultaneously account for geothermal inputs), but that seems beyond the scope of this manuscript.

*We thank the reviewer for this valuable comment. The fact that the correction using measured*
140  *rainfall values [e.g. Wai et al. 2008, Ezoe et al. 2002, Cheng and You 2010] gives negative*
*concentrations of some elements suggests that deep groundwater derived Chloride might also be*
*important (even 1% would significantly alter any correction) and therefore we have followed the*
*suggestion of the reviewer and removed our earlier correction. The dryness of the sampling period*
*means that this might stand as a source of error for some readers but we hope that the updated text*
145  *and argument will convince readers that the cyclic correction is likely smaller than the effect of*
*landsliding that we investigate. We have added & altered text to reflect this choice.*

*Lines 171-187: "**3.2. Cyclic Input***

*Solute input from atmospheric sources can be significant in rivers and streams, and is routinely corrected for in many*
*weathering studies. Previous work on the average chemistry of rainfall in different parts of Taiwan, has found that it is*
150  *highly dependent on location (e.g. higher pollutant content near the densely populated west coast (Wai et al. 2008) and*
*intensity of rainfall (typhoon rainfall boasts the highest solute concentrations in some cases (Cheng and You 2010)).*
*Using any of these measurements to correct our samples based on the assumption that all present chloride derives from*
*cyclic input (following e.g. Calmels et al. 2011) resulted in clearly erroneous estimates of the proportion of cyclic salts*
*in the water samples (e.g., >100% of dissolved $K^+$ and $SO_4^{2-}$ from cyclic sources). Notably, chloride is a minor*
155  *component in all samples, with a median value of 32 $\mu mol/l$ (minimum 14.6, maximum 175 $\mu mol/l$). This is a similar*
*range of values to rainwater measured in other coastal environments (Galloway et al. 1982; Nichol, Harvey, and Boyd*
*1997), and the overall correction of TDS is at most 10% even with the most concentrated rainfall we have found in*
*published literature (spot samples measured by Calmels et al. (2011) in the Taroko gorge). We note that deep*
*groundwater may also introduce chloride, which could lead to overestimates of cyclic input. Moreover, during the dry*
160  *sampling conditions, excess evaporation could lead to larger proportions of cyclic solutes. In view of the minor*
*amounts of chloride in our samples and because of the caveats mentioned above, we elect not to correct our samples for*
*cyclic input. In the discussion, we assess whether this simplifying assumption is valid."*

**Line 173 The authors are missing a citation here. Li et al. (2014; G3) also addressed the issues**
165  associated with the automatic mapping of landslides.

*We have added this reference; the Marc and Hovius (2015) paper explicitly addresses the effect of*
*amalgamation, but this is also part of the Li et al. (2014) work.*

170  ### Line 191 While it is true that the oxidation state of Si is +4 in natural waters, I don't think this
qualifies it as a cation. For example, most dissolved sulfur also has a positive oxidation state (+6),
but the authors correctly label it as an anion because it is speciated as sulfate in most natural waters.
Typically, Si is speciated as silicic acid in natural waters and, because the pKa of silicic acid is high,
it is fully protonated at the pH values observed in most rivers. So, maybe it is best to call Si a
175  neutral species.

*Both reviewer 1 and reviewer 3 have correctly pointed out our mis-description of Silicon as a*
*Cation. We have changed this throughout the manuscript.*

180  ### Line 200 The authors state that solute concentrations in their Taiwanese streams are high
relative to global rivers, but provide neither an average value nor a citation for the typical
concentrations of solutes in global rivers. Maybe the using values from the compilation of
Gaillardet et al. (1999 Chem. Geo.) would be useful here.

185  *The reviewer correctly points out that this claim is unsupported. We include both the suggested*
*citation as well as West et al (2005) who measured mountain streams.*

**Line 217 (and elsewhere) What is "weathering efficiency"? My interpretation would be that**

weathering efficiency is the same as what other authors call "weathering intensity", which is the
190   mass of rock solubilized by weathering divided by the mass of rock exposed by uplift. If this is
what the authors mean by weathering efficiency, then I do not understand how differences in solute
concentrations between streams and landslide seepage provide a useful constraint on it. Steams
presumably average over different timescales and have different mass fluxes compared to landslide
seepage, which could both obscure inferences about the proportion of rock solubilized. If the
195   authors mean something else, they should perhaps define this term in the text.

*The reviewer is correct that we have used weathering efficiency in the same vein as weathering*
*intensity, and we acknowledge that our lack of solute flux measurements make this an*
*unsubstantiated point. We have corrected this throughout, removing any use of the term efficiency.*
200
**Line 224 The authors argue that the effects of landslide area not evident in their data by**
reporting R2 values. First, what model are the authors using to describe this relationship? Using an
incorrect model could lead to a poor R2 despite there being an underlying relationship. Overall, this
would by much easier to assess if the authors showed us a scatter plot of landslide area and TDS.
205   After making one myself, its pretty clear that there is not an obvious correlation. Additionally, I am
a bit confused how the authors get negative R2 values. Unless the authors are calculating some sort
of adjusted R2 and just not telling us, they shouldn't be able to get negative values.

*We have added in a figure (fig 2) showing the relationship between the mapped landslide size and*
210   *the TDS in seepage from the deposit. In terms of the 'model' – we have clarified our phrasing to*
*refer directly to the correlation coefficient between area and TDS and volume and TDS.*
*Lines 308-310*: "In the Taimali catchment, there is no strong correlation between landslide size and TDS; $R^2$ values
for correlation between TDS and landslide area (Figure 2), and TDS and landslide volume are 0.11 and 0.09
respectively (p = 0.62, p = 0.70, respectively)."
215   *We have corrected the R2 values.*
*We have also included a discussion of how volumes were calculated.*
*Lines 221-225: "From the mapped landslide areas, we also estimate the volumes of individual slides, using global*
*area-volume relationships (Larsen et al. 2010). We lack direct measurements of the depth to which the landslides we*
*sampled have scoured. Errors in the mapped landslide area are assumed to be <20%, which are propogated into the*
220   *total volume estimated, as well as the catchment-wide volume estimates. This follows the approach described in*
*(Emberson et al. 2015), which allows direct comparison of the estimated landslide volumes in Taiwan and the WSA*
*(Figure 4)."*

225   ### Line 226
The authors report a correlation between sulfate and TDS and argue that this is evidence for the
effects of sulfide mineral oxidation. Such a correlation could arise for any number of reasons and is
observed globally (e.g., the data of Gaillardet et al.1999 Chem. Geo. show this relationship).
In general, solute concentrations are all negatively correlated with discharge, so variations in
230   discharge alone could produce a correlation between sulfate and TDS. Similarly, solute
concentrations should scale with water/rock interaction time, which could also give you a
correlation between TDS and sulfate. Dilution by rainfall, evaporative concentration, and end-
member mixing could all also give you a relationship between sulfate and TDS. And finally, as the

authors acknowledge, the dissolution of evaporites could also be responsible for this correlation.
To me, the fact that the catchment is underlain by marine shales is a strong argument
for sulfate being sourced from sulfide mineral oxidation. Personally, I do not see the additional
insight gained from reporting a correlation between TDS and sulfate concentrations, which is
expected (and observed) in most river systems independent of sulfide mineral oxidation. Personally,
I would find a scatter plot comparing elemental ratios much more convincing if the authors are
trying to make the point that sulfide mineral oxidation coupled to carbonate dissolution is the
dominant chemical reaction occurring within the landslide deposits. For example, SO4 /
(Na+K+Ca+Mg) versus Ca / (Na+K+Ca+Mg) would show what proportion of the cationic and
anionic budgets are sourced from the weathering products of coupled sulfide mineral oxidation and
carbonate dissolution.

*The reviewer is correct to point out that correlation between these two variables could result from
effects other than greater landslide input. We have replaced the Ca vs SO4 figure with two plots of
Ca/(Na+K+Ca+Mg) vs SO4/(Na+K+Ca+Mg) and Na/(Na+K+Ca+Mg) vs SO4/(Na+K+Ca+Mg),
which demonstrate that the increase in Calcium with respect to increasing Sulphate is not an
artefact of dilution or mixing. The fall in Sodium further reflects this. Looking at the original TDS-
SO4 plot in detail, the slope of the relationship is close to 2:1, which could well represent either
dilution or mixing of sources. As such, we remove this figure, and have rephrased the text to
emphasise the importance of the chemical ratios.*
*Lines 318-325:* "However, the ratio of calcium to the sum of other cations (sodium, potassium, magnesium and
calcium) is correlated with the ratio of dissolved sulphate to the same cations (Figure 3A; $R^2 = 0.64$, $p<0.005$), which
suggests a change in weathering regime. A decrease in the fraction of sodium to other cations also correlates with the
increasing fraction of sulphate (Figure 3B, $R^2 = 0.63$, $p<0.005$)."

**Line 235 The authors attribute the fact that solute concentrations are independent of landslide**
area to a heterogenous distribution of pyrite. Since landslide areas and volumes vary by 3 to 4
orders of magnitude respectively, does this imply that bedrock pyrite concentrations also vary by 3
to 4 orders of magnitude? Alternatively, could it be that bedrock pyrite concentrations are more or
less the same, but water fluxes co-varies with landslide area so that water/rock ratios are
approximately constant? Or, what if sulfide mineral oxidation is limited by the diffusion of oxygen
into the landslides? In this scenario, the lack of correlation between landslide area and sulfate
concentration arises from the fact that pyrite in deep landslide deposits is essentially inaccessible to
weathering. I suppose that this is what the authors mean by "the degree of its exposure for
oxidation", but it might be useful to develop this idea further.

*We agree with the reviewer. It is highly unlikely that the distribution of sulphides varies over several
orders of magnitude. The reviewer helps develop an idea which we briefly explored in Emberson et
al. NatGeo 2016, that the weathering boost may be limited in very large (deep landslides). In very
large Taiwanese landslides, oxidation may be limited; although we stress that size is unlikely to be
the only controlling factor on the diffusion of oxygen into deposits. The internal hydrology is likely
to be just as key, but we have no constraint on this.*
*Lines 337-340:* " The area and volume of landslides varies over 3 and 4 orders of magnitude respectively; it is
unlikely that the distribution of sulphides in the rock mass varies as much over short distance. Instead we suggest that
the absence of correlation between size and solute concentration is likely indicative of a limit to diffusion of oxygen in
landslides, either linked to the size or the unconstrained internal hydrology."

**Line 241 The say that landslides "remove the supply limit" on weathering. Conceptually, this does not make sense to me. To me, supply-limited weathering refers to conditions where all weatherable-material exposed by uplift is dissolved before being exported by erosion. This contrasts with kinetically-limited weathering, where un-weathered material is exported by erosion due to material residence times being shorter than the time required for complete reaction. Ferrier et al. (2016 G3) have a good discussion about these definitions. Based on these definitions, landslide weathering can be either supply-limited or kinetically-limited depending on how quickly landslide deposits are transported to the river channel and exported from the catchment. Since the authors do not constrain the residence time of landslides deposits with their study catchment, I am not sure how they are able to discuss supply- versus kinetically-limited weathering in landslide deposits.**

*The reviewer correctly points out that true 'supply limited weathering' would be the situation where all material is weathered before erosive export can occur. However, for the most reactive minerals, the amount of time for which they must be stored in the weathering zone to be completely dissolved is much less than the least reactive minerals. In a fresh landslide deposit, none of the minerals are depleted if they have been sourced from below the oxidation fronts; this means that weathering in the deposits is (at least initially) not limited by the absence of one or other mineral phase. However, our use of the term 'supply limit' invokes connotations of a more general, rather than mineral-specific, model, which was not our intention. We have removed the previous discussion of 'supply limits' to address this.*

**Line 246 Silicon, and to a lesser extent Ca, are not conservative in natural waters (e.g., Garrels and Mackenzie 1967, Jacobson et al. 2002 GCA, and many more), so I do not necessarily see how Si/Ca can be used as a reliable proxy for the proportion of cations sourced from silicate weathering. Furthermore, all of the reported data have elevated Na to Si ratios relative to common silicate minerals (e.g., albite), which is consistent with Si loss due to secondary silicate mineral precipitation (i.e. non-conservative Si behavior).**

*We observed significant precipitation of secondary calcite in the field in association with seepage from deposits, so we are well aware that Calcium is certainly not conservative. A full evaluation of this is well beyond the scope of this paper (in fact it forms part of a much longer manuscript we are currently preparing that investigates the $CO_2$ effect of landsliding), but the reviewer is correct in noting this. We don't have evidence of non-conservative Si behaviour other than the elevated ratios, but to account for this we have edited our discussion to focus on the Calcium:Sodium ratios rather than Silicon. As such, we use the ratio of Sodium to total Cations.*
*Lines 362-365: "the lower proportion of other purely silicate derived cations and silicic acid in landslide seepage (e.g. $Na^+$:$Ca^{2+}$ ratio always <0.5, compared to the silicate mineral value of 2.85 (Calmels et al. 2011)), as well as the markedly different behaviour of sodium and calcium (Figure 3A and 3B) supports the interpretation that weathering in the sampled landslides is dominated by the dissolution of carbonates."*

*We have removed the cyclic correction at the suggestion of the reviewer, which could lead to erroneous sodium in the measurements, but sodium in seepage is uncorrelated with chloride, suggesting this is not the case.*

**Line 255 While I think I agree with the authors, I find the wording in this section unclear. If I understand correctly, the authors are trying to argue that sulfate is sourced predominantly from insitu sulfide mineral oxidation as opposed to geothermal inputs. If one assumes that their single sample of geothermal water is representative of all geothermal inputs, then seepage Li concentrations can be used to place an upper bound on potential sulfate contributions from geothermal waters. Assuming all Li is sourced from geothermal waters, the observed Li enrichments in some of the seepage samples would be insufficient to account for a bulk of the sulfate budget due to the low SO4/Li ratio measured is in the single geothermal sample. So, to me, Li concentrations not "preclude" geothermal inputs, but instead constrain their contributions to the sulfate budget.**

*The reviewer is correct. Unfortunately we have not been able to sample geothermal water in other locations in the Taimali catchment, so have to rely on the single sample we obtained. We have changed the text to address the comment – the geothermal sample constrains the input of geothermally sourced sulphate to the overall sulphate budget, rather than the overall geothermal input.*

*Line 368-379: "Previous studies in Taiwan (Calmels et al. 2011) have shown that Sulphate is a key component in deep groundwater (sampled >100m below the surface), but clearly not all pyrite is oxidised at these depths. Groundwater circulating in the bedrock topography is unlikely to be the source of sulphate in seepage. Such water would not systematically surface at landslide sites, given the highly variable morphologic characteristics (size, scar/deposit ratio, position with respect to ridge crest and valley floor, and proximity to major faults) of individual landslides. The same argument applies to geothermal outflow. Moreover, hot spring waters sampled in the Taimali catchment have extremely high Lithium concentrations, 1610 µmol/l, and the lack of elevated Li$^+$ concentrations in the landslide seepage samples suggests the elevated sulphate in seepage derives from in-situ weathering. Other studies have found high concentrations of silicate-derived cations in deeper groundwater (Calmels et al. 2011); we do not observe any strong increase in the proportion of cations exclusively sourced from silicate minerals (particularly sodium – figure 3B) and sulphate in landslide seepage."*

**Line 258 I suppose this comment is similar to the one above. If one assumes that potential groundwater inputs would be characterized by high concentrations of both sodium and sulfate, then the limited co-variation between sodium and sulfate concentrations would suggest that variations in sulfate concentrations are not due to mixing with groundwater. While this seems like a reasonable argument, it'd be better supported with some scatterplots showing actual mixing relationships based on elemental ratios (not concentrations). The authors could even use literature data to better constrain some of the end-members.**

*We have taken the suggestion of the reviewer and introduced a scatterplot of SO4/(Ca+Mg+K+Na) vs Na/(Ca+Mg+K+Na) to demonstrate the negative correlation between the two in seepage samples.*

**Line 270 Does it matter that you are comparing solute concentrations from catchments with different surface areas?**

*We suggest that since all of the streams are saturated with respect to calcite (likely the key dissolving phase in the coupled sulphuric acid-carbonate weathering) the difference in size of catchment is likely of secondary importance. We also find no correlation between the catchment area and the concentrations of any of the dissolved species.*

**Line 276 If sulfide oxidation is limited by O2 diffusion, the weathering of silicates could be de-**

375    coupled from the oxidation of sulfides. In this case, a correlation between Na and sulfate would not be expected. And, as stated above, variations in Si concentrations are strongly influenced by secondary mineral precipitation which affects neither Na nor sulfate.

*We acknowledge that the inclusion of Silicon in this analysis made assumptions about the*
380    *conservative nature of Silicon in these waters, and have removed it. A decoupling of sulphide oxidation from silicate weathering may be possible, but we suggest that this requires invoking both oxygenated and deoxygenated (or at least without sulphate) hydrological pathways. This may well be the case, with landslides providing an oxygenation window for sulphides in catchments, and silicate weathering occurring elsewhere. We have changed the text to avoid confusion about what*
385    *we mean. Lines 379-383: "Extremely elevated $Na^+$ and $Mg^{2+}$ concentrations at some sites are not clearly associated with any other measured parameter, or with each other, and we have no ready explanation for these isolated observations at present. Since these high concentrations are not linked to dissolved sulphate, they may be linked to flowpaths with limited oxygen where sulphate oxidation cannot proceed, but we do not have a constraint on this."*

390    *This is also helped by defining the section in question as a results section, as suggested by reviewer 2 (see below).*
    *Lines 149-153: "**3. Methods** In this section, we report the analytical techniques used to obtain the data in section 4 summarise these results. In later discussion, we compare measurements of both chemical concentrations and ratios with physical data about landslides and catchments derived from remote sensing. In general, we focus on qualitative links*
395    *between these parameters, using statistical tests ($R^2$ coefficient of determination, and p-value to test significance of relationships) to support suggestions that parameters are connected."*

    ### Line 305 The authors state that the maximum solute concentrations observed in the landslides are 15 to 20 mM. While I don't disagree with this, I am not sure how this observation alone
400    provides insight into the "saturation" concentration of landslide weathering. While there probably is a maximum possible concentration, it need not be the highest concentration observed in this dataset. Also, saturation might be a poor word choice in this context since it has thermodynamic implications.

405    *We have rephrased this to remove saturation. The reviewer correctly points out that even if we haven't found higher concentrations, they may still exist.*
    *Lines 431-435: "The peak TDS values we measured in seepage are between 15000-20000μmol/l. Other locations may have greater concentrations, if local weathering has greater oxidation of sulphides, either due to more efficient distribution or greater diffusion of oxygen at depth in the landslide deposit. Such hypothetical, very high*
410    *concentrations, in excess of what we have recorded, would serve as an upper limit to solute concentrations in the entire catchment."*

    ### Line 319 The authors state that the amount of landsliding in each catchment is not correlated with the catchment slope. Where does this observation from from?
415

    *ArcGIS analysis of ASTER GDEM V2 (post Morakot) data provided the modal slope of each of the subcatchments. We have included this data in the table and included a description of the analysis in the text. Lines 226-227: "In addition, we have quantified the area, modal slope, and the mean and range in elevation of each catchment in an ArcGIS environment, analysing ASTER GDEM V2 data provided by NASA. These*
420    *data are reported in table 2."*

    ### Line 331 Lee et al. (2015 Env. Sci. Poll. R.) also report concentration-discharge relationships for Taiwanese rivers.

*We were not aware of this paper, and we thank the reviewer for the suggestion. It has been included in the text.*

**Line 336 Tipper et al (2006 GCA) report solute concentration-discharge relationship for Himalayan rivers and Torres et al. (2015 GCA) specifically link mountain catchments with "chemostasis" in the Andes/Amazon.**

*We have included these two references. We thank the reviewer for the suggestions.*

**Line 345 Again, I am not sure how a correlation between TDS and sulfate can be used to argue that solutes in stream water are sourced predominantly from landslides. This correlation is not unique to landslides and could arise from many different processes.**

*The reviewer is correct that in isolation this correlation can arise for other reasons. However, in these rivers that are saturated with respect to calcite, the effect of dilution is limited (especially at the dry conditions we sampled). This is supported by the comparison with sodium concentrations (Fig 5B). Sodium in streams does increase to some extent with chloride, suggesting a cyclic input. However, this is not correlated with the increasing sulphate concentrations. Figure 4C, showing increasing dissolved $SO_4^{2-}$ with increasing volume of landslides, supports our point that streams are mirroring the landslides. In these rivers the variability in TDS is not readily explained by catchment size, or other topographic variables. We feel it is important to point out that the sulphide oxidation not only affects the ratios of elements but also the overall concentrations.*

**Line 335 The statement "supersaturated everywhere with respect to Ca" doesn't make sense. Waters cannot be supersaturated with respect to an element. They can, however, be supersaturated with respect to a mineral phase.**

*We incorrectly referred to Calcium here. It has been changed to Calcite here and elsewhere.*

**Line 359 The authors define the term "non-equilibrium weathering" to describe landslide weathering because "the depletion rates of the various minerals involved are likely to be very different." I have a few issues with this statement. First, the fact that weathering occurs in the first place is because primary minerals are out of equilibrium with respect to surface conditions. So, by definition, all weathering is non-equilibrium. Second, chemical weathering in any setting will involve minerals that react at different rates. That said, I agree that landslide weathering is somehow "different". To me, the important difference is that landslides rapidly transport weatherable material from depth to the surface. In contrast, "hillslope" weathering involves the much slower transport of material from depth. Rapid transport of material from depth allows minerals that typically have deep weathering fronts (e.g., pyrite) to be exposed to weathering at the surface.**

*Our use of the term 'non-equilibrium' was perhaps overeager. We have altered the text to remove this phrase and replace it with a point emphasising that the weathering of all minerals occurs in 3 dimensions in deposits rather than at specific reaction fronts at depth in stable slopes.*
*Line 547-549: "The concept of a reaction front for weathering is less applicable to these settings, as the distribution of reactions will be controlled by the localisation of detrital sulphides and the flow paths providing oxygenated fluid."*

**Line 393 There definitely needs to be some citations for the links between weathering and atmospheric pCO2. Many of the papers cited elsewhere in the text should also be cited here. However, the authors could also consider citing:**

Lerman, A., Wu, L., & Mackenzie, F. T. (2007). CO2 and H2SO4 consumption in weathering and material transport to the ocean, and their role in the global carbon balance. Marine Chemistry, 106(1-2), 326–350. http://doi.org/10.1016/j.marchem.2006.04.004

Torres, M. A., West, A. J., Clark, K. E., Paris, G., Bouchez, J., Ponton C., Fakirs S.J., Galy, V., Adkins, J. F. (2016). The acid and alkalinity budgets of weathering in the Andes-Amazon system: Insights into the erosional control of global biogeochemical cycles, Earth Planet. Sci. Lett. http:/doi.org/10.1016/j.epsl.2016.06.012

*We thank the reviewer for the extra citations; the Torres et al. 2016 paper came out while this manuscript was in review. We have incorporated these and the Galy and France-Lanord (1999) reference used elsewhere in the paper. We mistakenly assumed that the $pCO_2$ link to weathering would be clear to the general reader and not need citation.*

**Line 400 The authors mention a "feedback between drawdown of CO2 and erosion." Since erosion rates are not necessarily modulated by atmospheric pCO2, I do not think this can be accurately called a feedback. Additionally, the authors should probably cite some of Maureen Raymo's work on this topic.**

*We agree with the reviewer; this is not a feedback (at least not a direct one). We adjusted the text, and have also cited the work of Raymo.*
*Line 590-592: "As erosion rates increase and mass-wasting through bedrock landsliding begins to dominate erosional budgets, the erosional impact on $CO_2$ and climate, achieved through sustained weathering of silicates (Berner and Kothavala 2001; Brady 1991, Raymo and Ruddiman 1992), will weaken."*

**Line 404 Should probably include the relevant citations for the links between sulfide mineral oxidation and erosion rates.**

*We have added in citations for Calmels et al (2007) as well as Torres et al. 2016, to address this.*

**Line 417 The authors state that the area affected by Typhoon Morakot is now a CO2 source but provide no evidence for this. The affect of weathering on atmospheric pCO2 depends on the precise balance between acid producing and acid consuming reactions (e.g., Torres et al. 2016 EPSL), which was not appropriately addressed in this manuscript.**

*This paragraph was discussed by all three reviewers. It has been extensively reworked to reflect these comments; the statement 'the area affected... is now a CO2 source' is unsubstantiated within this dataset. As mentioned above, we are currently finalising a manuscript dealing with this, and there was mistakenly some overlap between the trains of thought that went into the two separate pieces of work. As such, we now discuss the importance of quantifying substrate weathered and the acids at work to determine the CO2 output, but only suggest that it is possible that the area is now a source of CO2.*

*Lines 622-629:* "As a result, the area most affected by erosion due typhoon Morakot could be a net source of $CO_2$ to the atmosphere. The weathering boost is likely to end before the next major meteorological perturbation of southern Taiwan, so that the weathering of carbonates and silicates with Carbonic acid must be taken into account when considering the longer-term effects of erosion in South Taiwan. Future work to quantify the net sequestration or release of $CO_2$ would require a closer evaluation of the ratios of acid at work, as well as correction for the extensive secondary precipitation evident in the catchment."

**Line 448 It might be worth citing some of the other work on carbonate versus silicate weathering in New Zealand. For example,**

Jacobson, A. D., Blum, J. D., Chamberlain, C. P., Craw, D., & Koons, P. O. (2003). Climatic and tectonic controls on chemical weathering in the New Zealand Southern Alps. Geochimica et Cosmochimica Acta, 67(1), 29–46.

Moore, J., Jacobson, A. D., Holmden, C., & Craw, D. (2013). Tracking the relationship between mountain uplift, Silicate weathering, And long-term CO2 consumption with Ca isotopes: Southern Alps, New Zealand. Chemical Geology, 341, 110–127. http://doi.org/10.1016/j.chemgeo.2013.01.005

*The reviewer correctly notes the lack of citation here. We have incorporated the first of the suggested citations.*

**Line 455 Again, stochastic landsliding effects were previously incorporated into a weathering-erosion model by Gabet (2007 EPSL). This work should be mentioned and discussed here.**

*As with the introduction, we have incorporated the Gabet (2007) paper into the discussion.*

**Line 464 Again, the link between erosion and weathering the authors describe is not necessarily a feedback. The rates of tectonic uplift do not necessarily depend upon pCO2, so changes in weathering would not necessarily feedback on erosion rates. While I am aware that there are some possible weathering-erosion feedbacks (e.g., through rock strength and/or glaciation), I do no think that the authors are referring to these in this manuscript.**

*Here again we agree with the reviewer in general that this is not a direct feedback – we have clarified to incorporate the effect of climate on tropical storm intensity, which is what we were alluding to.*
*Lines 675-679:* "In this light, we draw attention to the possibility that disproportionate dissolution of highly labile mineral phases such as pyrite or carbonate as a result of landslide driven weathering may reduce the strength of the link between erosion and atmospheric $CO_2$ drawdown via silicate weathering in fast eroding settings, which in turn would modulate any feedback on erosion through changes in tropical storm intensity (e.g. Knutson et al. 2010). The role of active mountain belts in the global Carbon cycle remains ambiguous."

**Table 1. Was DIC measured directly? If so, how? If this column is actually charge balance, then it should be labeled as such. Charge balance does not necessarily equal DIC because of species like carbonic acid and organic bases. Also, it seems curious that sample TWS15-91 has no anions other than Cl and SO4 yet has a pH of 8.19. I tried to re-produce this pH with the reported solute concentrations using PHREEQC and couldn't get it to work. That said, I did not try very hard.**

Anyways, it might be worth looking into that sample a bit more to make sure there are not any analytical issues.

*DIC was assumed to be just HCO3- (at these pH conditions we feel this is a fair assumption) which*
570 *was calculated by charge balance. This is mentioned at the end of the methods section but we have clarified this and included this in the Table as well.*

*Having looked in more detail at the sample TWS15-91, there was indeed a problem with the analysis. I (RE) missed the fact that this sample was over the calibrated range in Calcium on the*
575 *initial OES run, and as such it was not rerun at a higher dilution. We have excluded it from the final analysis and manuscript.*

*We thanks Reviewer 1 for an extremely thorough and insightful review.*

580   **Anonymous Referee #2**

Emberson et al., Oxidation of sulphides and rapid weathering in recent landslides. Earth Surface Dynamics.
Determining how erosion and weathering are related to one another in tectonically active mountains
585   is important for understanding how mountain uplift influences geochemical cycling. Here Emberson et al. present geochemical data from surface waters in Taiwan. They focus on the chemistry of stream waters as well as waters that have flowed through landslide debris, deposited by extensive hillslope failures associated with Typhoon Morakot. The pyrite and carbonate bearing marine sedimentary rocks in the study area offer a mineralogical contrast to a complimentary study the
590   group conducted in the Southern Alps of New Zealand, where the mineralogy of the bedrock differs. Emberson et al. conclude that sulfuric acid generated by pyrite oxidation drives the dissolution of carbonate rock within the fractured landslide mass and that, above a threshold minimum of landsliding, that total dissolved solids, Ca, and sulfate increase with increasing catchment landslide volume. These results provide new, process-level insight on the generation of
595   solutes in tectonically-active landscapes, and will be of broad interest and is an appropriate contribution to Earth Surface Dynamics. I believe the manuscript can be accepted for publication following revision to address questions regarding several aspects of the paper.

*We thank the reviewer for taking the time to review our work. The comments have greatly aided its*
600   *development.*

Main comments: This manuscript lacks a distinct Results section, nor is the statistical treatment of the data clearly described in the Methods section, such that the reader does not know how the various dataset are analyzed with respect to one another until the statistics are presented as part of
605   the Discussion. A revised manuscript without an explicit Results section could work, but if so, additional description of the Methods are needed to better link the different sections of the paper and prepare the reader for the Discussion.

*We acknowledge that the methods and results sections are not clearly defined in the earlier version.*
610   *We have retitled our sections to better separate the results from discussion. We have noted how we have structured the results and discussion sections in the methods section.*
*Lines 146-153: "**3. Methods** In this section, we report the analytical techniques used to obtain the data in section 4 summarise these results. In later discussion, we compare measurements of both chemical concentrations and ratios with physical data about landslides and catchments derived from remote sensing. In general, we focus on qualitative links*
615   *between these parameters, using statistical tests ($R^2$ coefficient of determination, and p-value to test significance of relationships) to support suggestions that parameters are connected."*

Line 318-319. It is worthwhile to present data that demonstrate that the magnitude of landsliding (and TDS) does not scale with topographic metrics such as slope or relief. The reason being is that
620   erosion rates (and hence coupled weathering rates) scale with catchment topography. Hence it might be expected that, even in the absence of any landsliding, the range of TDS values could be explained by topographic variation of the watersheds. For the arguments presented in the paper, it is important to show this is not the case.

625   *The reviewer correctly signals that the topographic aspects of the catchments in question are missing, and we have fixed this by adding the modal slope and the relief of each of the catchments*

*(Table 2). The TDS values in the streams do not vary with any of these factors, and while we have not added a figure to demonstrate this lack of correlation, we discuss it in the text.*
*Lines 454-456:* "*Moreover, neither modal slope nor the range of elevation in a catchment is correlated with the measured TDS, suggesting these factors do not strongly control the weathering in this setting.*"

Line 406-414. It would be worthwhile to elaborate and make the logic very clear here, as this paragraph is laying out a conceptual framework, based on the observations, for understanding the role landslides play in the weathering budget of mountains. In particular, more explanation of the role of sediment retention time would be useful. The instantaneous weathering flux from the landslide debris should decay as a function of time, whereas the cumulative flux of weathering products from landslide debris will increase with time. Hence there should be more explanation to support the claim that long sediment residence times would have a limited effect on $CO_2$ budgets-if this indeed is the case, which seems unlikely. The return interval of landslide-triggering events should also play a role, as this ultimately sets the volume per time of landslide debris subjected to sulfuric acid weathering. A conceptual diagram showing the roles of residence time and return interval would be a useful approach to organize these thoughts in this paragraph. Additionally, there has been at least some work on 14C dating of alluvial deposits in southern Taiwan, presumably which record valley alluviation due to landsliding in some cases. Although these studies may not be in the same catchment as was investigated here, these studies could be drawn upon to estimate to a first order the length of time landslide derived sediment is likely to persist in the watershed.

Hsieh, Meng-Long, and Shyh-Jeng Chyi. "Late Quaternary mass-wasting records and formation of fan terraces in the Chen-yeo-lan and Lao-nung catchments, central southern Taiwan." Quaternary Science Reviews 29.11 (2010): 1399-1418.

Hsieh, Meng-Long, and Peter LK Knuepfer. "Synchroneity and morphology of Holocene river terraces in the southern Western Foothills, Taiwan: A guide to interpreting and correlating erosional river terraces across growing anticlines." GSA Special Paper (2002): 55-74.

*This is a really important point and one we want to address in detail. Material that is retained in the catchment will have a cumulative effect on weathering – as pointed out by the reviewer. However, if the labile minerals are depleted quickly, they will no longer impact the cumulative weathering after their depletion time. Silicate weathering can still occur in these deposits, and therefore affect $CO_2$, but the key physical effect of landsliding is to excavate the reactive phases. It is this landslide-driven effect that we were initially referring to, and we have tried to clarify our argument.*
*The terraces that are discussed in the two cited papers are sites we have visited; in an earlier version of this manuscript we had hoped to include the chemistry of seepage from these terraces as a long-lived version of the landslides, but the lithological differences between the terraces and landslides in the Taimali make a direct comparison difficult and the structure of the argument unwieldy.*
*As such, we have rewritten this paragraph to better summarise the key points. We hope that it is now clearer. The storage time of the material in catchments should have been included in the first place and we have incorporated both of the references to fix this.*
*Lines 564-579:* "*A second time scale of relevance is that of the physical removal of landslide debris from the catchment. If debris remains in a catchment after depletion of exposed pyrite, then its weathering will revert to mineral*

*reactions with Carbonic Acid. If the time to depletion is greater than the time required for removal of the debris, then the labile phases it contains will remain unweathered and will be removed through sediment transport processes. In the case of Taiwan, some samples of river bed material show Sulphur content of up to 1.07% (Hilton et al. 2014), suggesting that weathering in the landscape does not purge all sulphides brought to the surface by rock mass exhumation. In the opposite case the weathering from a given landslide will return to a constant background rate shared with other colluvium and soil covered parts of the landscape. How this scales up to the catchment scale depends on the return time of large drivers of mass-wasting, such as typhoons and earthquakes, and the subsequent advection of debris into the fluvial network and the transport capacity of this network. A disparity between short term production of sediment on hillslopes and the capacity for onward transport can lead to up to millenial residence times of event-produced sediment in mountain landscapes (Blöthe and Korup 2013). In the Taimali catchment, many headwater streams were still clogged with sediment 6 years after typhoon Morakot, and large colluvial fans had formed below several landslide-dominated tributaries. Elsewhere in Taiwan, mountain valleys contain large colluvial terraces, which have formed hundreds to thousands of years ago (Hsieh and Chyi 2010). This implies that time scales of pyrite depletion and of debris removal must both be considered in further explorations of the links between erosion and weathering in Taiwan."*

Lines 415-420. Here it would be useful to point the way forward in terms of what data are needed to determine the sign of the CO2 budget in a rapidly eroding mountain range. There are already elements of this in the paragraph, but a thorough explanation of what suite of measurements are needed would be a forward looking way to wrap up the manuscript. Additionally, with the understandably limited data that are available, is there at least a signal of what process is dominating the CO2 budget, sulfuric acid weathering of carbonate, carbonic acid weathering of silicates, or organic carbon burial? For example, back of the envelope calculations assuming regional-scale runoff values to infer fluxes for this study, and comparison with previous work on organic carbon export in Taiwan may be useful, even if imprecise. Even if a very first order budget cannot be calculated, due to lack of constraints in the values for the various parts of the budget, indicating that this is the case is a useful thing to do, as it points to the need for a different strategy in approaching erosion-CO2 links in mountains. On a similar note, the paper begins by asking a question regarding the general influence of landslides on weathering and roles lithology and climate may play (Line 42-44). In concluding the Discussion it would be worthwhile to return to the specific parts of that question and assess the state of knowledge, as informed by this work, and prior publications.

*As mentioned above, we have tried to rework this paragraph to remove the unsubstantiated claims and to explain which data we would need to discuss the CO2 impact quantitatively. We have also used the rewrite to actually answer the question that the reviewer correctly points out that we left unanswered in the earlier version.*

*Lines 622-629: "As a result, the area most affected by erosion due typhoon Morakot could be a net source of $CO_2$ to the atmosphere. The weathering boost is likely to end before the next major meteorological perturbation of southern Taiwan, so that the weathering of carbonates and silicates with Carbonic acid must be taken into account when considering the longer-term effects of erosion in South Taiwan. Future work to quantify the net sequestration or release of $CO_2$ would require a closer evaluation of the ratios of acid at work, as well as correction for the extensive secondary precipitation evident in the catchment."*

Line 427-429 (also line 241). There is a contradiction in this statement, in that the suggestion is that weathering is kinetically-limited, so long as the involved minerals are 'abundantly exposed'. The 'abundant exposure' is suggestive of supply limited weathering, and one interpretation of figure 3 (excluding the smaller landslide volumes in Taiwan), is that the supply of material by landsliding is setting the concentrations of dissolved species and likely also the weathering flux. My suggestion is

725 to be very explicit here regarding use of the terms kinetic and supply limited weathering. Pyrite oxidation depends on kinetics (as do all reactions), but when the numbers of minerals involved in such reactions are increased, likely by orders of magnitude due to fracturing of the rock mass during landslide runout, then the supply of minerals plays a very large role in setting the concentrations of various dissolved species measured in the water.

730 *Reviewer 2 echoes reviewer 1 in questioning our use of the terms kinetic and supply limited weathering. As mentioned in response to reviewer 1, we have rewritten the relevant sections to remove all confusion. Both reviewers have correctly understood our message, but more clarity is required for ease of understanding, and we have addressed this.*
*See comments above and lines 634-641: "Under these conditions, weathering is limited by the kinetics of the*
735 *reactions of the most labile mineral phases, but as long as these phases are abundant in the rockmass the rate of weathering is vastly increased. We have found that these labile phases may be minority constituents, driving the weathering in fresh landslide deposits away from the slower process affecting the surface materials elsewhere in the landscape over longer time scales."*

740 Other comments: Line 14. Insert 'landslide' before seepage

*Added 'landslide'*

Line 18 and elsewhere. Two significant digits on the R2 values is probably sufficient.
745
*Removed extraneous digits.*

Line 34. Several additional citations are warranted here, including Gabet and Mudd (2009), Ferrier and Kirchner (2008) and West, (2012). Also, Dixon and von Blanckenburg do discuss the role of
750 landsliding.

*See notes above in response to reviewer 1; added in extra citations.*

Line 38 and elsewhere. Many citations do not follow a uniform format, with author initials
755 appearing in some cases. Some editing the entries in the reference management software/database that was used here is needed.

*We thank the reviewer for their attention to detail! We have corrected this.*

760 Line 46. 'local' is ambiguous, perhaps replace with 'landslide'

*Local is indeed subjective. Changed it to 'landslide' accordingly.*

Line 56. 'importantly' can be omitted in this case and elsewhere in the manuscript. Alternatively,
765 switch the order of 'importantly' and 'propagate'

*Changed phrase to remove importantly.*

Line 79-81. Perhaps break this into two sentences or put parentheses around: either

770    directly from rainfall. . ...crest of landslides.

*Sentence removed and changed in response to comment from reviewer 1.*

Line 90-91. This statement ought to have a citation to support it.

775

*Added citation (Hartmann and Moosdorf 2012).*

Line 98. While I have not been to the field site, observations elsewhere in Taiwan would suggest, based on the continuity of vegetation coverage, that a soil (or mobile regolith) mantle, while not

780    necessarily laterally continuous, must cover a large fraction of the bedrock on hillslopes.

*We have added a citation in response to the comment of reviewer 1. Reviewer 2 is correct that the vegetation probably indicates more persistent soil than we had initially written. In the field we have struggled to get a good idea of what lies beneath the forest because of its density. Thus, we remove*

785    *the unsubstantiated assertion.*
*Lines 111-113: "Soils are generally only >1m in local concavities in the landscape (Lee and Ho 2009), but contiguous, dense vegetation on hillslopes that have not been dissected by landsliding suggests sufficient mobile regolith or soil is present to support flora."*

790

Line 105. Perhaps change 'in view of this' to 'in light of this'

*Changed to 'in light of this'.*

795    Line 115. The term 'channel fill' has an ambiguous meaning in this sentence. Does it have the same meaning as 'alluvium'? Some clarification is needed.

*Reviewer 2 correctly points out the ambiguity. We meant alluvium, and have changed it as such.*

800    Line 165. 'landslides' should be singular

*We respectfully disagree here, as we mapped many landslides.*

Line 169. Perhaps edit 'as prescribed by' to 'as described in'

805

*Changed to 'described in'.*

Line 173. A G-cubed paper by Gen Li and colleagues also provides an important example of the effects of amalgamating landslide areas. Li, Gen, et al. "Seismic mountain building: Landslides

810    associated with the 2008 Wenchuan earthquake in the context of a generalized model for earthquake volume balance." Geochemistry, Geophysics, Geosystems 15.4 (2014): 833-844.

*Thanks to reviewers 1 and 2 here for this suggestion. We have included this reference.*

815    Line 197. Edit so that 'sit at' is removed

*We decided that this sentence was extraneous and it has been removed.*

Line 201. The statement that TDS values are high with respect to global concentrations needs a
820  citation.

*Added citations in response to comments from reviewer 1 and also this comment. (Gaillardet et al.
1999, West et al. 2005)*

825  Line 220-221. Whether deeper landslides will expose proportionally more calcite to weathering
relative to shallow landslides will depend on the depth of the calcite weathering front. Whereas
much calcite is likely lost from the soil, the soils are likely thin (<1m?) and a small proportion of
the total landslide thickness. Given the lithology there may well be considerable calcite in the near-
surface environment in the study sites. Hence the lack of a correlation between TDS and landslide
830  size may not be altogether unexpected.

*This is another good argument for the lack of correlation between size and the seepage chemical
concentration. We have adjusted our argument to incorporate this point.*
*Lines 299-305: "In the WSA, TDS concentrations in landslide seepage are strongly impacted by the dissolution of*
835  *trace calcite, the abundance of which is likely to be greater at depth (Brantley et al. 2013). The depth of a landslide
scales with its area (Larsen et al. 2010), implying that the amount of calcite as a proportion of the total landslide
volume correlates with the landslide size. This correlation will be limited to the range of landslides where a significant
proportion of excavated material is from above the depth to which calcite is depleted. By contrast, in the largest
landslides, near-surface materials, poor in calcite, form only a small proportion of the deposit, and a correlation with*
840  *size my no longer hold."*

Line 223. If reduced hydraulic conductivity at the base of landslide debris, relative to debris that is
stratigraphically higher, has been documented this statement needs a citation. Otherwise some
clarification is needed.
845

*Here we have changed the text to be more exact; larger slides may have reduced diffusion and
advection of fluids and oxygen to the deeper parts due to the non-linear area-volume scaling. We
cite our previous work as a reference for the linearity of the landslide area to water input as in that
work we also calculated how the area of landslides scales with the area that drains into them.*
850  *Lines 305-307: "Moreover, large landslides may also be subject to reduced diffusion of fluids and oxygen to the
base of the deposits; the amount of water entering the slide scales with the area of the slide (Emberson et al. 2015), but
the volume increases at a greater rate (Larsen et al. 2010)."*

Line 224. Replace the slashes with 'per'.
855

*Are correlations reportable as e.g. 'TDS per landslide volume'? We have changed the phrasing to
'$R^2$ values for correlation between TDS and landslide area, and TDS and landslide volume...'*

Line 236. To link this with the statistics above, replace 'size' with 'area or volume'
860

*This sentence has been removed, but we have used 'area and volume' in the latest version in
accordance with this comment.*

Line 241. Avoid starting a sentence with 'This'. Instead explicitly indicate something to the effect of: 'Generating landslide debris from unweathered bedrock removes the supply limit. . .'

*We removed this sentence to respond to a comment from reviewer 1.*

Line 242. Replace 'and surrounds' with some explicit definition of what is meant by surrounds. The other sites in southern Taiwan that were sampled?

*Corrected to 'other sampled sites in Southern Taiwan'*

Line 253. Omit importantly. Unless there is evidence the groundwater is 'circulating', simply refer to it as 'deep groundwater'

*Removed 'importantly' and 'circulating'.*

Line 271. Replace 'tops' with 'exceeds'

*Replaced.*

Line 302. Rainfall is not dependent on area, but runoff is. Clarify as needed.

*Sentence removed to address comment from reviewer 1. However, we agree with reviewer 2 here, our initial statement was not accurate.*

Line 316. '..and these may be a source of variability. . .' It is unclear what 'these' refers to in this sentence and this sentence doesn't really follow from the previous one or set up the following one well. Some editing is needed here.

*Edited the sentence – replaced 'these' with 'these factors, such as local slope or hillslope length,...'*

Line 321. 'importantly' can be omitted.

*Removed 'importantly'*

Line 329. First two words have unclear meaning.

*Changed 'Since' to 'As', but we are unclear about what is unclear?*

Line 334. Replace 'in keeping' with consistent

*Changed to 'consistent'.*

Line 336. Problems with these references. e.g., should be Lyons et al.

*Thanks to the reviewer for pointing this out. Changed accordingly throughout the manuscript.*

910     Line 348. A 'likely' should be inserted between 'exhumation' and 'not'. There are no data on the distribution of sulfide minerals within the rock, so you can only hypothesize that this variability also controls the variability in seepage concentrations at individual landslides.

*Added 'likely'.*

915

Line 357-358. Suggest editing to: '. . .and the associated generation of sulfuric acid.'

*Adjusted to the suggested phrasing.*

920     Line 360. Different than what? Please complete this thought.

*Sentence removed in response to comment from reviewer 1.*

Line 361. Is there a better term than 'minority' minerals – reactive, highly soluble, labile? Choose
925   one and use it consistently throughout. For example, there are likely many phases present in low abundance that do not contribute to non-equilibrium weathering.

*The reviewer is correct to point out that minority is an overly general term. We have replaced it where relevant. We have used the words 'reactive' or 'labile' to be as specific as possible. Where the*
930   *word minority is used it is now explicitly to point out that these reactive phases are often a low proportion of the lithology.*

Line 368. Some sort of punctuation to separate the sample numbers and the values of the measurements would be helpful.

935

*Added punctuation.*

Line 376. The statement regarding weathering by carbonic acid dominating in other parts of the landscape should be qualified. It is likely the case, but data are not discussed that support this. Can
940   the data from the rivers be used to support this inference?

*Removed statement '[reactions which] dominate elsewhere in the landscape.'. If weathering doesn't happen via sulphuric acid, carbonic acid is likely the key agent of weathering (e.g. Torres et al. 2016, Galy and France-Lanord 1999). However, we have no support for the statement that this*
945   *dominates elsewhere in the landscape.*

Line 378. It is not clear what 'restricted' means in this context. Please clarify.

*Replaced 'restricted' with 'some samples of river bed material' – we had meant that we only have*
950   *one reference for the sulphur content of bed material.*

Line 381. There is a comma that can be omitted.

*Removed.*

955

Line 403. Elaborate on '. . .a more central role,. . .' It will be useful to be explicit regarding what the central role refers to.

*Altered to 'Weathering of carbonates will form a greater portion of solute budgets'*

960

Line 406. Suggest replacing the 'As' with 'when'

*Replaced as with when.*

965 Line 426. Again, unclear if the waters in the landslide debris indeed circulate.

*Replaced 'persistently circulating' with 'infiltrating'*

Line 430. Omit 'in the first instance' or replace with another term, such as foremost.

970

*Replaced 'in the first instance' with 'primarily'*

Line 434. Replace 'units' with 'rocks' and replace 'This' with a more explicit description of the process.

975

*Replaced 'units' with 'rocks'. Expanded 'This' to 'This oxidation'.*

Line 436. Meaning of 'subsumed' is unclear in this context. Some editing is needed.

980 *Changed phrase to 'while the weathering of silicate minerals is not elevated'.*

Line 339. Omit 'first' or replace with another term than does not imply a temporal relationship.

*Replaced 'first' with 'better correlated'*

985

Line 454. 'mineralogical' instead of 'lithological'?

*Changed 'minor lithological' to 'reactive mineral'*

990 Line 457. Gabet 2007, EPSL is a study that has examined the stochastic role of landslides in weathering.

*Added Gabet citation in response to this comment and comment from reviewer 1.*

995 Figure captions. 'Total dissolved load (TDS)' – the acronym should match the first letters of each word, e.g., total dissolved solids or TDL.

*Replaced Dissolved 'load' with dissolved 'solids'*

1000 Figure 1. It would be useful to have an inset map showing Taiwan with a box showing the area of

Figure 1a. This inset could go in the lower right corner of Fig. 1a.

*Added inset figure.*

1005    Figure 2. There should be at least two labels on the y-axis in each panel to easily allow the reader to see the data range.

*Figure 2 has been altered to address comments from reviewer 1, but in doing so we also added the extra y-axis labels.*

1010

*We thank reviewer 2 for a thoughtful and detailed review.*

**Anonymous Referee #3**

The authors argue that landslides place unweathered material at the land surface, exposing pyrite to oxidation which drives weathering. They discuss many implications. The paper was interesting and this phenomenon deserves attention because it is important. I have seen dissolved Fe emanating from beneath landslides in Puerto Rico (and soon thereafter precipitating), right after a landslide, so I know this happens and the central idea is not surprising. Pyrite oxidizes fast when exposed to oxygen and water and especially when bacteria are involved. So that is not that surprising. I think what the authors are trying to do that is interesting is quantify this effect by looking at different watersheds with different densities of landslides. This is an interesting idea. But I don't think they quite nailed down their story. I have a hard time even knowing exactly what they did show, and what this means exactly. I suggest that the authors work hard to probe their data more. With that effort, their results will be more impactful.

*We thank the reviewer for taking the time to put together a thorough and careful review of our manuscript. It is encouraging to hear of similar observations in field sites we have not visited! We have not intended to suggest that this process should come as a surprise. What is novel about this study is the empirical data from seepage and the demonstration of the effect on a large scale. Apart from our work on weathering in landslide deposits in New Zealand (Emberson et al. 2015) we are not aware of other work on this subject.*

Part of the problem is that much of the phrasing is a bit vague and confusing. I got tired trying to figure out what they meant. Even the abstract has such phrasing. For example, this phrasing in the abstract is awkward and unclear and should be revised: "Bedrock landslides create conditions for weathering where all mineral phases in a lithology are initially unweathered within landslide deposits, and therefore the most labile phases dominate the weathering at the outset and during a transient period of depletion." I can't really read the abstract and get the main point. Given that I am not surprised about pyrite oxidation being accelerated by landslides. . .what is the big take-home message that they can show? One interesting aspect that is not explained clearly is why Taimali operates differently than WSA: shouldn't this basic difference be in the abstract?

*We apologise for the complex phrasing in parts! Where possible, we have tried to address this, including the sentence in question in the abstract. The big difference between the WSA and the Taimali – the importance of pyrite – has been included.. We are encouraged by the fact that the reviewer is not surprised by our conclusions; this means that our observations have wider significance. Line 20-23: "The predominance of coupled carbonate-sulphuric acid driven weathering is the key difference between these sites and previously studied landslides in New Zealand (Emberson et al. 2015), but in both settings increasing volumes of landslides drive greater overall solute concentrations in streams."*

I do think the authors should rework their paper before final publication. Here are some points of various importance to consider:

1. The key question in my mind is the depth to which landslides scour the landscape in comparison to the depth of the pyrite depletion zone and the carbonate depletion zone. The authors sort of mention this but not fully. Where are the reaction fronts for pyrite and carbonate in these rocks? And what is the depth to which landslides scour? Often pyrite is oxidized down to the water table.

*The reviewer is exactly correct to pinpoint this relationship; how deep do landslides scour in comparison to the depth of the pyrite and carbonate depletion zone? The answer is we don't have a good idea. The depth of landsliding does scale with area (Larsen et al. 2010) but this can vary over several orders of magnitude globally. We therefore have only very limited constraint on the depth to which the landslides we measure actually scour. The exception to this is that we only assess bedrock landslides, meaning they excavate at least below the saprolite-bedrock interface. However, in the mountains of Taiwan there exists limited data on the depth of soils, and we are not aware of any systematic study investigating the depth of pyrite or carbonate depletion. The lack of boreholes in the mountains of Taiwan or the Taimali in particular (Taiwan Water Resources Agency Hydrological Yearbook(s) 1990-2012) means we don't have a good sense of the depth of the water table either. The heterogeneity of flow paths in mountain belts (e.g. Andermann et al. 2012 Nature Geoscience, Calmels et al. 2011 EPSL) means that the depth might well vary widely.*
*To address these points, we have expanded the discussion of the scour depth vs depletion depth in the initial discussion (where we reiterate the points above). See lines 241-249, and also lines 299-303.*

2. Please discuss the calculation of bicarbonate by charge balance. This would usually have a large or reasonably large error just because it is a small difference between larger numbers. But, for example, the calculation could also have error due to neglect of organic anions in the waters, or any other anions. What is the error in bicarbonate? (and it should not be labelled DIC, it should be labelled bicarbonate). I see that the measurements of sulfate and chloride are each +_10%...seems like the calculation of bicarbonate by difference might have big error bars. In that regard, there are most likely too many significant figures on the bicarbonate (and probably some of the other element concentrations?)

*- The error in Bicarbonate is given in the table, but we have added it to the text. The calculated error – the root mean square error of the uncertainties on the other elements – is actually small compared to the value of 10% that we use, because the small analytical error on Calcium tends to dominate the overall error. We estimate a 10% error based on previous work (Galy and France-Lanord 1999 GCA) that demonstrated charge-balance estimates are generally within 10% of calculated alkalinity. Lines 165-166: "Bicarbonate ($HCO_3^-$) was calculated by charge balance, which has been shown to introduce errors of approximately 10% (Galy and France-Lanord 1999)."*

*-Naturally, alkalinity is not just HCO3-. We make the simplifying assumption that at the pH values measured, it is the only important Carbonate species (compared to an error of 10%) and that the alkalinity derived from other components is also negligible. Lines 166-168: "We make the simplifying assumption that the Dissolved inorganic carbon (DIC) is formed only of $HCO_3^-$, which is generally applicable at the range of pH values measured (Zeebe and Wolf-Gladrow 2001)."*

3. The calculation of bicarbonate was then used to assess calcite supersaturation. I would think there is a big error in this calculated SI as well.

*- To our knowledge, PHREEQC does not automatically calculate the uncertainties on the saturation indices. We expect that manual calculation of every possible saturation index for the range of chemistry defined by all errors (i.e. each variation of errors for each chemical element) would require an extreme amount of time; we cite previous work (Moore et al. 2013, Chemical*

1105 *Geology) where saturation indices are also published without errors attached. Nevertheless, we acknowledge the problems with publishing data without errors attached; any guidance the reviewer could offer as to how to calculate these errors would be welcomed.*

4. I wouldn't capitalize element names (Calcium versus calcium).

1110

*Corrected capitalisation where necessary.*

5. Generally, I would not call Si "cationic load" although we do usually refer to it as a cation simply because we analyze it with other cations. But it is present as a neutral species unless the pH is very

1115 high. For this reason, Si does not contribute to cation balance.

*The reviewer echoes reviewer 1 here, and we have adjusted both text and figures to address this issue. Silicon is no longer referred to as a cation, and we have removed it from figure 3 (replaced with sodium).*

1120

6. With respect to Sr: what does this mean? "peaks at 30 umol/L

*Replaced 'peaks at' with 'the maximum measured value of which is'*

1125 7. The paper has a lot of assertions that are really undefended assumptions. Such assertions need to be defended more thoroughly: "the depth of a landslide scales with the area". . .is this always true for all locations? or was Larsen reporting about one locality more than another? I am also not sure about "the proportion of calcite available correlates with the size of the issuing landslide". I think this statement could be true but doesn't have to be true. Are there data corroborating this

1130 assumption? The authors later state that "the lack of correlation between landslide size and seepage concentration suggests that the distribution of pyrite is highly heterogeneous.." I am not sure I would say the pyrite distribution is heterogeneous but I would argue that in many places there could be a pyrite oxidation front like those identified in Chigira et al. 1990; 1991; Drake et al., 2009; Brantley et al., 2013. I suppose that the base of landsliding in any given area might be somehow

1135 correlated with the depth to pyrite, but I think it is also very possible that the depth of landsliding might not correlate with the depth to pyrite.

*We refer back to the comments we made on the first point by reviewer 3; the Larsen dataset is global, but it does vary over several orders of magnitude. In response to both this comment and*

1140 *earlier comments from reviewer 1 we have adjusted the argumentation in this section, as the reviewers correctly point out that the proportion of carbonate in the deposit will depend on more factors than just the size. Again see lines 241-249, and also lines 299-303.*

8. The paper needs some mineral abundance data and element abundance data.

1145

*In terms of mineral abundance data, we have incorporated data from Kao et al. (2004) suggesting 0.05-0.27% Sulphur in South Taiwan Sediment sections. Carbonate abundance is estimated to be between 0.28-0.66% (Hilton et al. 2014) although this does not sample the Taimali. Elemental abundance data have been measured by Selveraj and Chen (2006), but we have not included this*

1150 *citation as we have not explicitly used elemental abundance data in the manuscript.*

9. The authors start using the phrase "Landslide weathering efficiency" without defining it. What exactly is this?

1155 *Our use of the term efficiency has caused problems. We have addressed this above and it has been removed to avoid confusion.*

10. Please explain why you are using Kendall's tau value?

1160 *We initially used K-Tau values to estimate the degree of dependence of the variables; this does not make the same assumptions about linearity and Homoscedasticity as the R2 coefficient, but is (in our experience) not as widely used. However, in general the trends which we observe are linear and the assumption of homoscedasticity is qualitatively valid, so we have removed any discussion relating to K-Tau to avoid confusion.*

1165

11. Line 243, acidity means the base neutralizing capacity of a water. Do the authors mean acidity or protons? (Pyrite is a source of protons.)

*The reviewer is correct; acidity is the wrong term to use here. We have changed this to state: 'this*
1170 *provides a source of sulphuric acid'. Elsewhere we have also changed 'acidity' to 'acid' as appropriate.*

12. I suggest the authors include some sort of figure or more text to defend this statement: "the lower proportion of purely silicate derived cations in landslide seepage . . .supports the
1175 interpretation that weathering in the sampled landslides is dominated by the dissolution of carbonates".

*The reviewer correctly points out that we have not substantiated this point. We cite previous work in Taiwan (Calmels et al. 2011) for the generally used ratio of calcium to sodium in silicate rock*
1180 *(0.35), which is well below the ratio in the dissolved load (>>1) that we measure to support the importance of carbonate weathering in landslides. Line 363: "(seepage (e.g. $Na^+:Ca^{2+}$ ratio always <0.5, compared to the silicate mineral value of 2.85 (Calmels et al. 2011)),"*

13. I don't understand this: ". . .a significant proportion of the sedimentary pyrite contained within
1185 the rock mass survives exhumation. . ." I think in general, pyrite moves up and out of the system like any other mineral until it gets oxidized at some depth related to the rate of erosion and the amount of water. Is this is what is meant by survival (death by oxidation)? It is generally oxidized down to the water table unless it is a very pyrite rich rock.

1190 *We broadly agree with the interpretation of the reviewer, but point out that much of the pyrite contained within solid rock mass is inaccessible to circulating groundwater. In general we expect that a fraction of the pyrite will be oxidised in deep groundwater (e.g. Calmels et al. 2011) but some survives to be exploited by landsliding. We have rephrased this sentence to reflect this. The depth at which groundwater was collected by Calmels and coauthors (100s metres below the surface) is not*
1195 *systematically excavated by the landslides we sample (if at all). Lines 368-369, and more importantly 612-614, where we state cite work showing that some pyrite is removed in river*

*sediment:* *"Shorter sediment residence times promote the export of unweathered sediment to the ocean and its burial in marine basins. This can result in a fraction of sulphides contained in the rock mass bypassing the weathering window (Hilton et al. 2014),"*

1200

14. The authors argue that Na and Si do not vary with sulfate but Ca and K do vary with sulfate. They argue that this points to multiple pathways relevant for silicate weathering on a catchment scale. I don't see how an observation about landslide seepage can tell us about the other pathways: it can only tell us about the landslide contribution, not the other contributions. I think there are

1205    nested reaction fronts. . .deeper flowpaths will interact with parent lithology and shallower flowpaths will not. The authors are arguing that landslides bring some of the deeper material to the surface: landslides are egg beaters. What we need to know is, how deep do they scour relative to where the minerals are?

1210    *Again the reviewer here stresses the need for information on how deep the landslides scour relative to the reaction fronts. We point out that in such a tectonically dissected mountain belt with extensive faulting and fracturing (e.g. Lin et al. 2008 Engineering Geology) there might well be fluid percolation that is not necessarily 'top down'. This may limit the extent to which 'nested' reaction fronts can persist on a catchment scale. Nevertheless, our phrasing has evidently led to confusion.*

1215    *When we say there are multiple pathways for silicate weathering then we mean that the landslides are not responsible for all of the silicate weathering, but they may have a role to play in some of it. In other words, there is a landslide pathway, and an indeterminate number of other pathways, which we haven't constrained. See paragraphs between lines 359-388.*

1220    15. I think this is unsupported as written: "the relationship between landslide volume and stream solute concentration will also saturate above a hypothetical maximum concentration"

*This sentence was also pointed out to be unsubstantiated by reviewer 1, and we have rephrased it to address both this comment and the comment of reviewer 1. Lines 431-435:* *"The peak TDS values we*

1225    *measured in seepage are between 15000-20000μmol/l. Other locations may have greater concentrations, if local weathering has greater oxidation of sulphides, either due to more efficient distribution or greater diffusion of oxygen at depth in the landslide deposit. Such hypothetical, very high concentrations, in excess of what we have recorded, would serve as an upper limit to solute concentrations in the entire catchment."*

1230

16. Waters cannot be supersaturated with respect to an element. . .only wrt a mineral (unless we are talking about native sulfur or gold or some such). See line 356. This needs to be fixed.

*Thanks to reviewers 1 and 3 for picking up on this. It has been changed accordingly (to 'with*

1235    *respect to calcite'.) (line 270).*

17. I don't understand this sentence at all! I think it should be deleted or re phrased. "We describe this style of weathering as non equilibrium weathering as the depletion rates of the various minerals involved are likely to be very different."

1240

*This sentence has caused issue for the other reviewers (see above) and as a result has been removed and the discussion rephrased.*
*Lines 550-553:* *"Landslide weathering rates will remain elevated until the relevant reactive mineral(s) have been*

*depleted, after which weathering in landslides is likely to proceed much like in other parts of the landscape, driven by organic and carbonic acids. The initial abundance of trace sulphide and/or carbonate together with their exhaustion, over longer time scales, is likely to be an important control on the duration of rapid weathering in landslides."*

18. What is important is not the "minority minerals" but "reactive minerals". Once the reactive minerals are removed, then the landslide material will weather like all the other weathered material in the landscape (i.e. material above the reaction front for the reactive mineral). . .unless the landslide is grinding up the material and increasing surface area, which is another possible effect. I do think that pyrite is often a minor mineral but that is not what is important about it.

*This echoes comments from the other reviewers; we have rephrased sections of the manuscript that refer to minority minerals to instead refer to reactive minerals.*

19. Once pyrite oxidizes, weathering will revert from H2SO4 to carbonic plus organic acids.

*Presuming the reviewer here means that we've not mentioned the organic acids, we have added them into the discussion. Lines 550-552:* "Landslide weathering rates will remain elevated until the relevant reactive mineral(s) have been depleted, after which weathering in landslides is likely to proceed much like in other parts of the landscape, driven by organic and carbonic acids."

20. The authors' use of "ambivalent" is incorrect: "the impact of landsliding on climate is ambivalent". I think the authors mean ambiguous.

*Changed to 'ambiguous'.*

21. I am not sure that the authors can defend the statement, "the area most affected by erosion due [to] typhoon Morakot is now a net source of CO2"

*As pointed out by the other reviewers, we have overstated our case here. We have rephrased the paragraph to only state substantiated claims or, where we are not certain, to state this uncertainty. See whole paragraph – lines 581-629.*

22. Conclusions: I don't understand this sentence: Five years after landsliding, carbonic acid is a weathering agent of lesser importance and the weathering of silicate mienrals is subsumed.

*This sentence has been rephrased in response to this comment and a comment from reviewer 2. (Subsumed is the wrong term here). Line 644-645:* "Five years after landsliding, carbonic acid is a weathering agent of lesser importance, while the weathering of silicate minerals is not elevated."

23. The term "lithological phases" is not correct. A lithology is a lithology and a phase is a phase. . .not sure what a lithological phase is. . .a mineral?

*This sentence has been rephrased in response to this comment and a comment from reviewer 2. (Lines 656-657)*

24. In figure 4, SiO2 is silica, Si is silicon.

*Figure 4 has been changed to replace Si with sodium.*

25. The way the figure on page 20 is plotted is confusing. The fonts on the y axis differ above and below the break. I don't really understand what the tick marks mean on the y axis, because of the
1295    break in the axis. Why plot the two figures together in this way?

*We've altered this figure to remove any confusion over the overlap, as well as adding extra labels to the y-axis. To answer the question posed by the reviewer, we plotted these together originally to use space as efficiently as possible, but we've changed this to improve the clarity.*
1300
*We thank reviewer 3 for a review that reflects extensive field and lab experience, which has helped us improve our work.*

[revised manuscript text omitted]

All samples are shown uncorrected for either cyclic or hydrothermal input.

** Charge-balance calculations suggest no DIC, meaning a calcite saturation index is not possible to calculate.

[Figure]

[Figure]

Fig.3A

[Figure]

[Figure]

Fig.4

[Figure]

Fig.5